# STYXL1 regulates CCT complex assembly and flagellar tubulin folding in sperm formation

Yu Chen [1,2,4], Mengjiao Luo[1,4], Haixia Tu[1,3,4], Yaling Qi[1,4], Yueshuai Guo[1], Xiangzheng Zhang[1], Yiqiang Cui [1], Mengmeng Gao[1], Xin Zhou[1], Tianyu Zhu[1], Hui Zhu[1], Chenghao Situ [1]✉, Yan Li [3]✉ & Xuejiang Guo [1]✉

Tubulin-based microtubule is a core component of flagella axoneme and essential for sperm motility and male fertility. Structural components of the axoneme have been well explored. However, how tubulin folding is regulated in sperm flagella formation is still largely unknown. Here, we report a germ cell-specific co-factor of CCT complex, STYXL1. Deletion of *Styxl1* results in male infertility and microtubule defects of sperm flagella. Proteomic analysis of *Styxl1^-/-* sperm reveals abnormal downregulation of flagella-related proteins including tubulins. The N-terminal rhodanese-like domain of STYXL1 is important for its interactions with CCT complex subunits, CCT1, CCT6 and CCT7. *Styxl1* deletion leads to defects in CCT complex assembly and tubulin polymerization. Collectively, our findings reveal the vital roles of germ cell-specific STYXL1 in CCT-facilitated tubulin folding and sperm flagella development.

Male infertility arises from spermatogenic defects and is mainly characterized by oligo-astheno-teratozoospermia. Asthenozoospermia, which has abnormal sperm motility, is a common cause of male infertility[1]. The sperm flagella contain three parts including midpiece, principal piece and end piece, and provide a structural basis for sperm motility[2]. It contains a central axoneme with '9 + 2' peripheral doublet microtubules and central microtubule pair (CP)[3] surrounded by accessory structures including mitochondrial sheath (MS), outer dense fibers (ODFs) and fibrous sheath (FS)[4]. Defects in the axoneme structure can cause morphological abnormalities of the flagella and severe sperm motility disorders[5]. Mutations in components of '9 + 2' microtubule structure are shown to cause multiple morphological abnormalities of the flagella (MMAF), a subtype of severe asthenozoospermia with various morphological defects of the sperm flagella (absent, short, coiled, bent or irregular width)[6–10].

Microtubules are the core components of cilia and flagella axoneme[11], and are also required for spindles in cell division[12].

Microtubules are formed by alpha- and beta-tubulin, which must be folded by chaperonin containing TCP-1 (CCT) complex prior to microtubule assembly[13]. In eukaryotic cells, the chaperonin CCT complex consists of eight different subunits (CCT1 to CCT8) and forms two rings that are stacked back-to-back in a specific arrangement (CCT1-3-6-8-7-5-2-4)[14]. In the open form, tubulin interacts with the N- and C-terminal of the CCT subunits in an amorphous state. In the closed form, tubulin is located in the folding chamber, interacting extensively with the apical domains of the CCT subunits[15]. The tubulin folding by CCT complex is regulated by co-factors in microtubule assembly. In *drosophila*, Misato can occupy the CCT complex central cavity and is required for proper assembly of spindle microtubules[16]. However, the regulation of tubulin folding by CCT complex in flagellar formation has not been well studied.

Protein phosphatases are known to play regulatory functions in diverse cellular processes, such as signal transduction, metabolism, subcellular trafficking, and inflammation, via phospho-signaling

[1]Department of Histology and Embryology, State Key Laboratory of Reproductive Medicine and Offspring Health, Nanjing Medical University, Nanjing 211166, China. [2]Medical Research Center, Changzhou Maternal and Child Health Care Hospital, Changzhou Medical Center, Nanjing Medical University, Changzhou 213000, China. [3]Department of Clinical Laboratory, Sir Run Run Hospital, Nanjing Medical University, Nanjing 211166, China. [4]These authors contributed equally: Yu Chen, Mengjiao Luo, Haixia Tu, Yaling Qi. ✉e-mail: situchenghao@njmu.edu.cn; yanli@njmu.edu.cn; guo_xuejiang@njmu.edu.cn

networks or non-phosphorylation pathways[17–20]. Moreover, phosphatases are also involved in sperm flagella development. In *Chlamydomonas*, protein phosphatases PP1 is a part of the central pair and PP2A is anchored on the doublet microtubules to control flagellar motility[21]. Protein phosphatase 4 (PPP4) deficiency causes sperm tail bending defects and poor sperm motility in mouse[22]. However, the molecular mechanism of phosphatase in the regulation of sperm flagella remains to be further explored. Pseudophosphatase STYXL1 is a member of PTP (protein tyrosine phosphatases) superfamily, but lacks the critical histidine and nucleophilic cysteine residues vital for catalysis[23]. In somatic cells, STYXL1 is reported to inhibit G3BP-1-induced formation of stress granules[24], to promote apoptosis by reducing PTEN-like mitochondrial phosphatase (PTPMT1) activity[25] and to induce outgrowths in PC-12 cells[26]. However, the role of STYXL1 in spermatogenesis of germ cells still remains unknown.

Here, we systemically analyzed the expression of phosphatases in spermatogenesis, and found that STYXL1, as a testis-abundant pseudophosphatase not previously studied in testis, is essential for male fertility. We found that deletion of *Styxl1* led to MMAF. *Styxl1−/−* mice showed defects of microtubule organizations in sperm flagella but normal spindle structure in meiotic metaphase. STYXL1 interacted with CCT complex, and acted as a co-factor for the assembly of CCT complex, which folds tubulins in sperm axoneme formation.

## Results

### STYXL1 is a germ cell-specific pseudophosphatase in testis

Protein phosphatases are key regulators of many cellular processes and diseases[27]. To investigate the role of protein phosphatases in testis, we first analyzed their expression patterns. Among 233 phosphatases[28] 23 were testis-enriched[29,30] (Fig. 1A). According to da Cruz et al.'s expression profile of testicular cells[31], most phosphatases were highly expressed from pachytene spermatocytes, and ten phosphatases had highest expressions in spermatids (cluster 2 in Fig. 1B). Spermiogenesis of spermatids is a highly complex process during which haploid round spermatids undergo a series of programmed transitions and eventually transform into spermatozoa[32]. Further literature analysis of these ten phosphatases (PPM1B, STYXL1, PGP, EYA4, CDC14A, PPEF1, DUSP21, DUSP13, PP2D1, and HDHD1A) showed that (Fig. 1C) three phosphatases, including PPM1B, PGP and EYA4, are essential to prevent embryonic lethality[33–35]; deletion of *Cdc14a* causes spermatogenic defects[36]; DUSP13 is involved in meiosis regulation;[37] PPEF1 is related to sperm fertilizing and motility[38]; PP2D1 is dispensable for spermatogenesis[39]; HDHD1A may be central in the epigenetic and genomic regulation of sex chromosome aneuploidies[40]. Only STYXL1 and DUSP21 have not been studied before in mouse, while STYXL1 is highly conserved from *elegans* to *humans* according to multiple sequence alignment analysis (Supplementary Fig. 1A).

STYXL1 belongs to the dual specificity phosphatase (DSP) family, and is defined as pseudophosphatase on account of alterations of catalytic motifs to catalytical inactivation[28]. To validate the expression profile of *Styxl1*, we performed RT-PCR in eight adult mouse tissues, and the results showed that *Styxl1* was most highly expressed in testis (Fig. 1D). Kit^{w/wv} mice have no germ cells except only undifferentiated spermatogonia and normal somatic cells in testis[41]. We performed qRT-PCR analysis of *Styxl1* in Kit^{w/wv} and control testes, and found no expression in Kit^{w/wv} testis (Supplementary Fig. 1B). During the first wave of spermatogenesis, postnatal 1-week, 2-week, 3-week, 4-week, 5-week and 7-week testes show development of germ cells to spermatogonia, early spermatocytes, round spermatids, elongating spermatids and sperm, respectively. qRT-PCR analysis showed that *Styxl1* had significantly increased expression in 3-week testis and it achieved the highest expression from week 4 (Fig. 1E). We further purified Sertoli cells, pachytene spermatocytes, round spermatids, elongating spermatids and mature sperm and found that *Styxl1* was expressed in pachytene spermatocytes, round and elongating spermatids and

sperm but not in Sertoli cells (Fig. 1F), consistent with its absence of expression in Kit^{w/wv} testes. Altogether, these results indicated that STYXL1 is a germ-cell-specific pseudophosphatase in testis, and it may play a vital role in spermiogenesis of spermatids.

### *Styxl1* deletion leads to male infertility and MMAF

To examine the functions of *Styxl1* in spermatogenesis, we generated *Styxl1* knockout mice using CRISPR-Cas9 system with sgRNA targeting exon 5 (Fig. 2A), and obtained two independent *Styxl1* mutant lines with deletion of 4 bp and 74 bp, respectively (Supplementary Fig. 2A, B). To evaluate the deletion of STYXL1 protein in mutant mice, relative quantification of STYXL1 in testis was evaluated using targeted protein quantification based on a parallel reaction monitoring (PRM) method by LC-MS/MS. We found that STYXL1 protein was completely absent in the testes of both two mutant lines (Fig. 2B, Supplementary Fig. 6), thus the mutant mice with larger deletion of 74 bp were mainly used in the in-depth functional analysis, and presented as *Styxl1−/−* mice subsequently. *Styxl1−/−* mice were viable while fertility test showed that *Styxl1−/−* male mice were completely infertile (Fig. 2C). Compared to the wildtype (WT) control mice, the testis/body weight ratio was observed with no difference in *Styxl1−/−* male mice (Fig. 2D). Moreover, the number of sperm collected from *Styxl1−/−* mice cauda epididymis was extremely reduced (Fig. 2E), and computer-assisted sperm analysis (CASA) showed that the sperm motility and sperm progressive motility also decreased in *Styxl1−/−* mice (Fig. 2F, G, and Supplementary Movie 1). To investigate the etiology of *Styxl1−/−* mice, we performed the histology analysis of epididymis by hematoxylin and eosin (H&E) staining. The results showed that the epididymis of WT mice was full of sperm, while fewer sperm were observed in the epididymis of *Styxl1−/−* mice (Fig. 2H). Moreover, sperm from cauda epididymis in *Styxl1−/−* mice exhibited severe morphological defects including abnormal heads, bent tails and coiled tails (Fig. 2I, J). The analysis of *Styxl1^{Δ4bp/Δ4bp}* male mice showed the same phenotypes as *Styxl1−/−* mice (Supplementary Fig. 2C–I, and Supplementary Movie 1). Taken together, *Styxl1* deletion results in reduced sperm motility and increased morphological abnormalities of sperm accounting for male infertility.

### *Styxl1−/−* sperm show microtubule disorders

To analyze the spermatogenic defects in *Styxl1−/−* testis, we performed PAS staining of *Styxl1−/−* testis and found that elongating spermatids showed nuclear malformation since step 10 (Fig. 3A), and had fewer flagella in the lumen at stage VII-VIII. The zoom-in views of malformed elongating spermatids and corresponding schematic diagrams were shown in Fig. 3B. And *Styxl1^{Δ4bp/Δ4bp}* male mice also showed defects in nuclear malformation in elongating spermatids and fewer flagella in the lumen at stage VII-VIII, which are the same as those in *Styxl1−/−* male mice (Supplementary Fig. 3A, B). Because the *Styxl1−/−* elongating spermatids showed malformed nuclei, we analyzed the nuclear condensing by staining transition protein 1 (TNP1). *Styxl1−/−* spermatids showed similar expression of TNP1 from step10 to step15 compared to the WT spermatids, suggesting that nuclear condensation was not affected after deletion of *Styxl1*(Supplementary Fig. 3C). Previous reports showed that manchette is a microtubular platform between the perinuclear ring surrounding the nucleus and the elongated sperm axoneme, and plays an important role in the nuclear shaping and sperm flagella formation during spermiogenesis[42]. We performed immunofluorescent staining of α-TUBULIN, and found that the manchette was elongated abnormally along the nuclear surface in *Styxl1−/−* elongating spermatids (Supplementary Fig. 3C). We further analyzed ultrastructural changes of elongating spermatids undergoing spermiogenesis using transmission electron microscope (TEM). The *Styxl1−/−* spermatids exhibited severe ultra-structural defects, including abnormally constricted nuclei at the site of perinuclear rings, deformed acrosome and malformed nuclei, and asymmetric and abnormally elongated manchette microtubules (Fig. 3D).

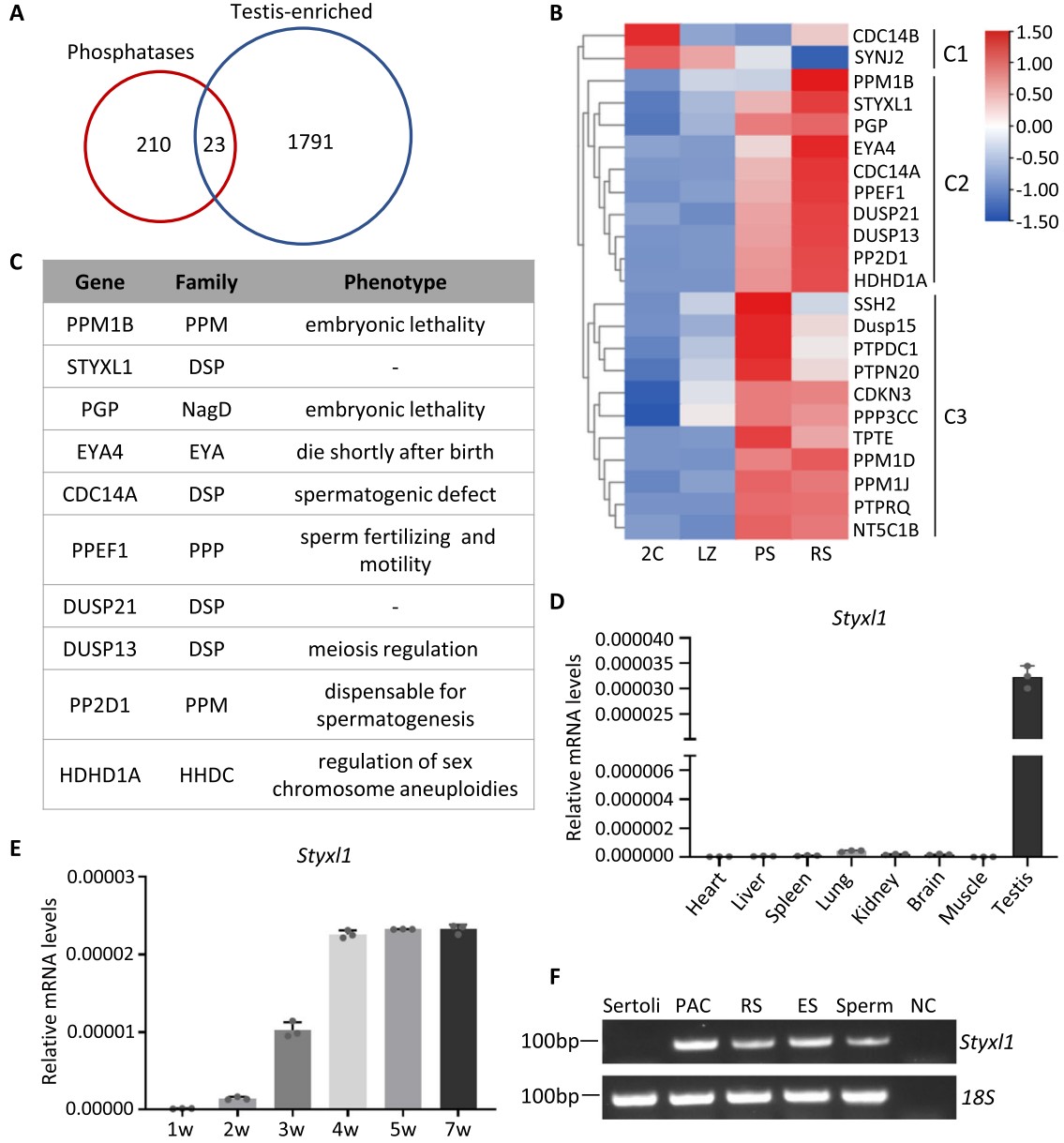

**Fig. 1 | Expression of pseudophosphatase *Styxl1* in mouse testis. A** Venn diagram showing overlap between known phosphatases and testis-enriched genes. **B** Heatmap of expression levels of testis-enriched phosphatases in different developmental cells. 2C: somatic cells and spermatogonia, LZ: leptotene/zygotene spermatocytes, PS: pachytene spermatocytes, RS: round spermatids. **C** The phosphatase family and phenotypes of proteins from Cluster C2 with highest expression in round spermatids. **D** qRT-PCR analysis of expression levels of *Styxl1* mRNA in eight mouse tissues (*n* = 3 mice per group). **E** qRT-PCR analysis of expression levels of *Styxl1* mRNA in different postnatal weeks of mice testes (*n* = 3 mice per group). **F** Expression level of *Styxl1* mRNA in different types of spermatogenic cells by qRT-PCR. 18S rRNA was used as a loading control. PAC, pachytene spermatocytes; RS, round spermatids; ES, elongated spermatids; NC, negative controls. Data are presented with the mean ± SD. Source data are provided as a Source Data file.

Given that deletion of *Styxl1* led to reduced sperm motility and sperm flagella defects, we further examined flagellar formation during spermatogenesis using AC-TUBULIN immunofluorescent staining. The results showed an apparent scarcity of sperm flagella visible at stage VII-VIII in *Styxl1^-/-* mice compared with WT mice (Fig. 3D), indicating that STYXL1 might be involved in the sperm flagella formation. Next, we observed flagellar formation from round spermatids to elongating spermatids during spermiogenesis by AC-TUBULIN immunofluorescent. The results showed that flagellar formation was abnormal from step 9 when round spermatids began to deform and elongate (Fig. 3E). Furthermore, corresponding with above findings, mature sperm from cauda epididymis in *Styxl1^-/-* mice also showed coiled or bent flagella by immunofluorescent staining of AC-TUBULIN (Fig. 3F). We then analyzed the ultra-structures of the sperm flagella using TEM

(Fig. 3G), and found that *Styxl1^-/-* sperm showed displacement of ODF and MTs or absence of CP in the midpiece, abnormal '9 + 2' structure or absence of CP in the principal piece, and lack of MTs in the end piece. The abnormal axoneme structure of sperm flagella observed in *Styxl1^-/-* mice revealed that STYXL1 plays a critical role in the development of sperm flagellar axoneme.

### *Styxl1* deletion leads to abnormal expression of sperm flagellar proteins

*Styxl1* deletion led to sperm flagellar axoneme abnormalities. To analyze the relationship between STYXL1 and sperm axoneme, we overexpressed HA-STYXL1 in NIH3T3 cells and analyzed the co-localization between STYXL1 and primary cilia by staining AC-TUBULIN. The results showed that STYXL1 was localized in

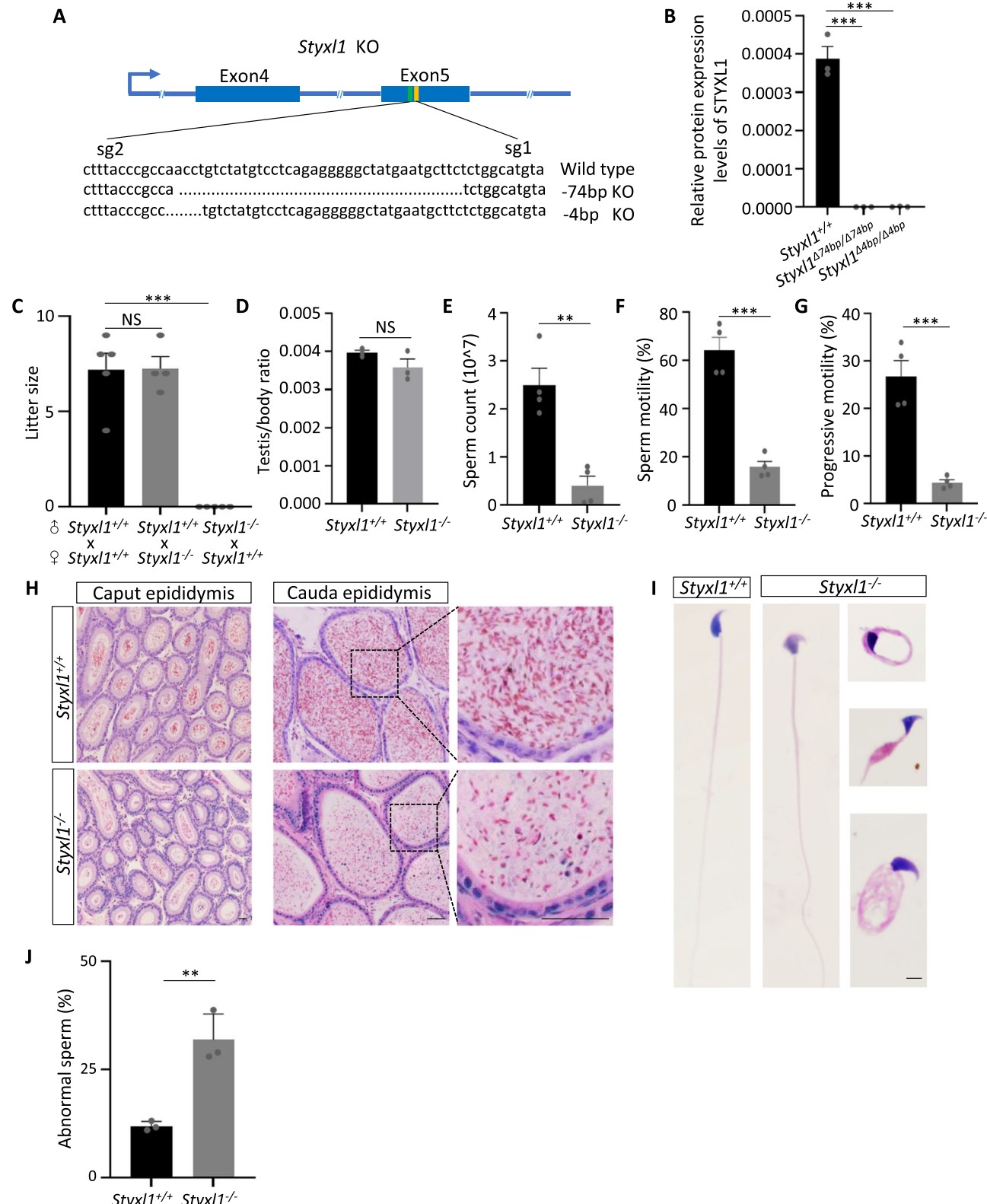

AC-TUBULIN-positive cilia (Fig. 4A). To study the regulatory roles of STYXL1 in cilia functions, we knocked down endogenous *Styxl1* using shRNA, and measured the length of induced cilia in NIH3T3 cells. qRT-PCR analysis validated knockdown efficiency for two shRNAs that targeted different sites of *Styxl1* (Supplementary Fig. 4A). After serum starvation, *Styxl1* knockdown greatly decreased the length of cilia in NIH3T3 cells while overexpression HA-STYXL1 synonymous mutations at siRNA binding region

(HA-STYXL1<sup>syn</sup>) in knockdown cells can rescue the shortened cilia (Fig. 4B, C). Thus, STYXL1 is important for cilia formation.

In order to investigate the molecular mechanism of STYXL1 in the regulation of sperm flagella, sperm from *Styxl1*<sup>+/+</sup> and *Styxl1*<sup>-/-</sup> mice were collected for quantitative proteomics analysis using TMT-based liquid chromatography-tandem mass spectrometry (LC-MS/MS) approach. A total of 6429 proteins were quantified, and 1517 proteins ($p < 0.05$, Foldchange > 2) were differentially expressed (Supplementary Fig. 4B

**Fig. 2 | Phenotype analysis of *Styxl1* knockout male mice. A** Schematic diagram of targeting strategy by CRISPR/Cas9 and the knockout alleles in the two founder lines. **B** Quantification of relative expression levels of STYXL1 protein in adult *Styxl1$^{Δ74bp/Δ74bp}$* and *Styxl1$^{Δ4bp/Δ4bp}$* testes by PRM (*n* = 3 mice per group). ***p* < 0.0001 using one-way ANOVA followed by Dunnett's multiple comparisons test. **C** Litter size of adult *Styxl1$^{-/-}$* male and *Styxl1$^{-/-}$* female mice (*n* = 5 mice for *Styxl1$^{+/+}$* female group, *n* = 4 mice for *Styxl1$^{-/-}$* female group). NS *p* = 0.9977 and ***p* < 0.0001 using one-way ANOVA followed by Dunnett's multiple comparisons test. **D** Statistics analysis of adult *Styxl1$^{+/+}$* and *Styxl1$^{-/-}$* testis/body weight ratio (*n* = 3 mice per group). NS *p* = 0.1670 using two-tailed Student's t-test. **E–G** Quantitative analysis of sperm count (***p* = 0.0020 using two-tailed Student's t-test) (**E**), sperm motility (****p* = 0.0002 using two-tailed Student's t-test) (**F**) and progressive motility (****p* = 0.0006 using two-tailed Student's t-test) (**G**) of adult *Styxl1$^{+/+}$* and *Styxl1$^{-/-}$* mice (*n* = 4 mice per group). **H** H&E-stained caput epididymis and cauda epididymis of *Styxl1$^{+/+}$* and *Styxl1$^{-/-}$* mice. Scale bar:50μm. **I, J** The morphologies of *Styxl1$^{-/-}$* sperm by H&E staining (**I**) and the percentage of sperm abnormalities (***p* = 0.0046 using two-tailed Student's t-test) (**J**) in *Styxl1$^{-/-}$* cauda epididymis compared with controls (*n* = 3 mice per group). Scale bar:5μm. NS, not significant; **, *p* < 0.01; ***, *p* < 0.001. Data are presented with the mean ± SEM. *n* = 3 biologically independent samples were included in each group (H, I). Source data are provided as a Source Data file.

and Supplementary Data 1). Among these differential proteins, 986 proteins were downregulated (65%) and 531 proteins were upregulated (35%) in *Styxl1$^{-/-}$* sperm (Fig. 4D). Further gene ontology analysis of downregulated proteins showed that biological process terms such as "cilium movement", "cilium organization", "sperm motility" and "axoneme assembly" were enriched (Fig. 4E), and cellular components terms such as "motile cilium", "sperm midpiece", "axoneme", "dynein complex" and "microtubule" were significantly enriched (Fig. 4F). Among these proteins, we found that tubulins including α-TUBULIN (Tuba3a) and β-TUBULIN (Tubb3, Tubb4b) were significantly downregulated in *Styxl1$^{-/-}$* sperm. These results were consistent with above findings of aberrant sperm flagella in *Styxl1$^{-/-}$* mice, suggesting that STYXL1 may regulate the expression of axoneme proteins in the development of sperm flagella.

### STYXL1 interacts with and regulates the expression of the chaperonin CCT subunits

To clarify how STYXL1 regulated axonemal protein expression, we analyzed its interacting proteins by GST pull-down assay followed by LC-MS/MS analysis using a GST-MBP-fused mouse STYXL1 protein (Supplementary Fig. 5A). We found that CCT complex containing CCT1-CCT8 subunits were identified (Fig. 5A). CCT complex is essential for the folding of key proteins driving diverse cellular processes, including cytoskeletal proteins, tubulin[43–45]. To validate the proteomic findings and identify direct binding partners, we used the yeast two-hybrid system (Y2H) to analyze the interactions between STYXL1 and CCT1-CCT8 proteins. As shown in Fig. 5B, we found that CCT1, CCT6 and CCT7 interacted directly with STYXL1, SUN1C and KASH5 were used as positive controls[46]. STYXL1-interacting chaperonin CCT complex may regulate the folding and stability of sperm flagellar proteins during sperm flagella formation.

STYXL1 possesses a rhodanese-like domain in N-terminal segment and a C-terminal tyrosine-protein phosphatase domain (Fig. 5C). To examine which region in STYXL1 mediated the interactions between STYXL1 and CCT1, CCT6 and CCT7, we analyzed protein domain relationships using co-immunoprecipitation (co-IP) assays in HEK293T cell line. The results showed that full length STYXL1 interacted with CCT1, CCT6, and CCT7 while their reciprocal interactions were independent of the tyrosine-protein phosphatase domain (Fig. 5D, E). In MAPK phosphatases (MKPs), also members of DUSPs, the N-terminal rhodanese-like domain can bind mitogen-activated protein kinase (MAPK) through its kinase interaction motifs (KIM) to downregulate the kinase activity and hence play an important role in controlling cellular responses and the physiological outcome of signaling[47]. However, the KIM motif of the rhodanese-like domain in STYXL1 protein is mutated and lacks consecutive critical arginines[48], which are required for MAPK docking[49]. Our study showed that MAPK was not identified in the proteomic analysis of co-immunoprecipitated proteins of STYXL1, indicating a possible different function of rhodanese-like domain of STYXL1 from that of MAPK kinase. Further yeast two-hybrid experiments and reciprocal co-immunoprecipitation analysis showed that the N-terminal rhodanese-like domain mediated the interactions between STYXL1 and CCT1, CCT6 and CCT7, which were independent of the tyrosine-

protein phosphatase domain, indicating important functions of the rhodanese-like domain in DUSPs.

To identify the roles of CCT1, CCT6, and CCT7 in sperm flagella, we separated adult mouse sperm into Triton X-100 soluble, SDS soluble, and SDS resistant fractions. SLC2A3, AC-TUBULIN and AKAP4 were detected as markers for Triton-soluble, SDS-soluble and SDS-resistant fractions, respectively[50]. Western blot analysis showed that CCT1 and CCT7 localized exclusively in the SDS resistant fraction, and CCT6 is mainly localized in the SDS resistant fraction (Fig. 6A). We also performed immunofluorescent staining analysis, and found that CCT1, CCT6, and CCT7 colocalized with AC-TUBULIN along the flagella of mouse sperm. Moreover, we have found that CCT1, CCT6 and CCT7 remained restricted to the sperm flagella but showed discontinuous distribution after deletion of *Styxl1* (Fig. 6B). We next examined expression of these proteins in *Styxl1$^{+/+}$* and *Styxl1$^{-/-}$* sperm, and found that the expression level of CCT1 significantly decreased in *Styxl1$^{-/-}$* sperm (Fig. 6C), indicating CCT complex may be involved in sperm flagella abnormalities after *Styxl1* deletion. To study the function of CCT complex in flagella formation, we analyzed the effects on the length of primary cilia in NIH3T3 cells by knocking down endogenous *Cct1* (Supplementary Fig. 5B). The results showed that the cilia length decreased significantly after induction by serum absence compared with the controls, and the inhibitory effects of *Cct1* siRNA can be rescued by overexpression of FLAG-CCT1 synonymous mutations at siRNA binding region (FLAG-CCT1$^{syn}$) (Fig. 6D, E). Therefore, STYXL1 regulates sperm flagella biogenesis by its interacting CCT complex.

### STYXL1 is required for CCT Complex assembly and affects sperm microtubules organizations

Above findings indicated that *Styxl1$^{-/-}$* sperm showed defective microtubule structures. The full CCT complex is built by two rings comprising eight different subunits[51], and can correctly fold α-TUBULIN and β-TUBULIN to polymerize in microtubule formation in spindle assembly during mitosis[16]. It is not known if it can also participate in the tubulin polymerization in sperm axoneme formation. To characterize the roles of STYXL1 in CCT complex assembly and tubulin folding. We performed gel filtration assay in wildtype and *Styxl1* knockdown NIH3T3 cells. With the systemic analysis of all the 8 subunits of CCT octamer complex, we found that, in the elution fraction 8 of wildtype cells, all the 8 CCT complex subunits and α-TUBULIN and β-TUBULIN were co-eluted, indicating that tubulins are folded by intact 8-subunit CCT complex (Fig. 7A). However, in *Styxl1* knockdown NIH3T3 cells, CCT1 was not co-eluted with α-TUBULIN and β-TUBULIN, and shifted to later fractions with smaller molecular weights. CCT2 and CCT4 were also shifted to elution fractions of smaller molecular weights in *Styxl1* knockdown NIH3T3 cells. These results indicated that the formation of CCT octamer complex containing all 8 subunits was compromised in *Styxl1* knockdown cells. Previous studies have shown that CCT proteins can form micro-complexes with less than 8 CCT subunits[52]. It seems that STYXL1 is essential for the assembly of intact CCT octamer complex with all 8 subunits to fold α-TUBULIN and β-TUBULIN. To analyze the roles of STYXL1 in tubulin polymerization in microtubule assembly, we performed in vitro MT sedimentation assay to examine the polymerization state of α-TUBULIN and β-TUBULIN. We found that the vast

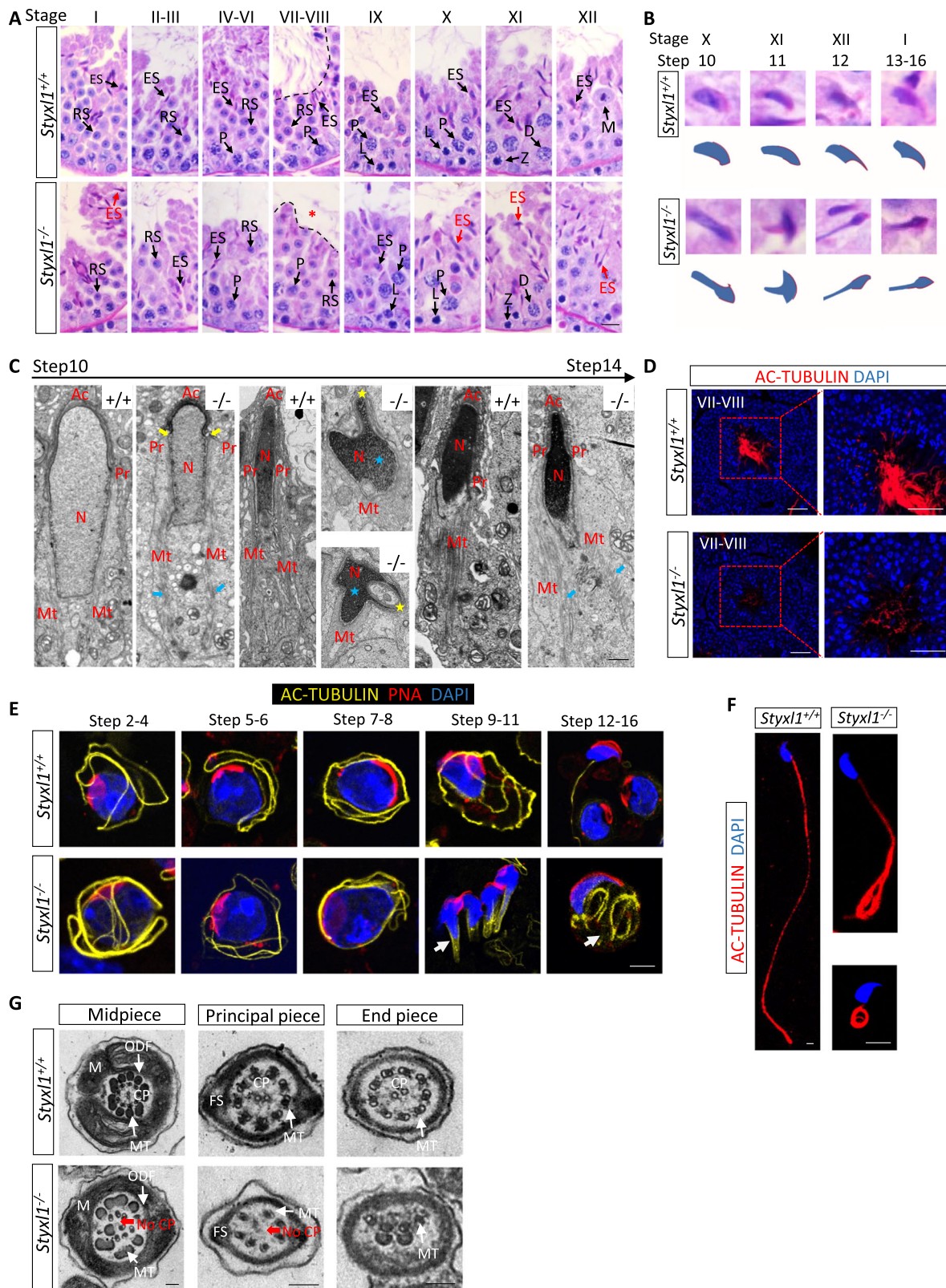

majority of α-TUBULIN and β-TUBULIN were able to pellet under polymerization conditions in the control group. In contrast, in *shStyxl1* treated groups, α-TUBULIN and β-TUBULIN remained in the supernatant and were unable to polymerize (Fig. 7B, C). We further analyzed the expression and polymerization of tubulins in *Styxl1⁻/⁻* sperm. We found that α-TUBULIN and β-TUBULIN were significantly decreased in *Styxl1⁻/⁻* sperm lysate (Fig. 7D, E), which was consistent with our above

quantitative proteomic results of *Styxl1⁻/⁻* sperm (Fig. 4F). The MT sedimentation assay further showed that *Styxl1* deletion led to defects in tubulin polymerization in sperm (Fig. 7F, G). Collectively, STYXL1 acts as a co-factor for CCT complex assembly in the tubulin folding/polymerization pathway in sperm.

Spermatogenesis undergoes mitotic, meiosis and spermiogenesis to form mature sperm. *Styxl1* is expressed from pachytene

**Fig. 3 | Spermiogenesis defects in *Styxl1⁻/⁻* mice. A** Different stages of seminiferous tubules in PAS-stained *Styxl1⁺/⁺* and *Styxl1⁻/⁻* testes. The red arrow indicates abnormal nuclei of elongating spermatids. Asterisk indicates defects of sperm tails. L, leptotene; Z, zygotene; P, pachytene; D, diplotene; RS, round spermatid; ES, elongated spermatid. Scale bar: 25μm. **B** Enlarged pictures of different steps of *Styxl1⁺/⁺* and *Styxl1⁻/⁻* elongated spermatids pointed by arrows in (**A**) were presented. Schematic diagrams were denoted at bottom. **C** Ultrastructural analysis of *Styxl1⁺/⁺* and *Styxl1⁻/⁻* spermatids by transmission electron microscopy. The yellow arrows indicate abnormal constrictions at perinuclear rings. The blue arrows indicate asymmetric and abnormally elongated manchette microtubules. The malformed nuclei (blue asterisk) and acrosomes (yellow asterisk) were also indicated. Ac, acrosome; Pr, perinuclear ring; N, nucleus; Mt, manchette. Scale bar: 1μm. **D** Immunofluorescence of AC-TUBULIN (red) in stage VII-VIII *Styxl1⁺/⁺* and *Styxl1⁻/⁻* seminiferous tubules. Nuclei were stained with DAPI (blue). Scale bar: 20μm. **E** Different developmental steps of spermatids isolated from *Styxl1⁺/⁺* and *Styxl1⁻/⁻* testes were stained using AC-TUBULIN (yellow), PNA (red) and DAPI (blue). Arrow indicated abnormally coiled flagella. Scale bar: 10μm. **F** Immunofluorescence analysis of *Styxl1⁺/⁺* and *Styxl1⁻/⁻* sperm stained by AC-TUBULIN (red) and DAPI (blue). Scale bar: 5μm. **G** TEM analysis showing the cross-section of midpiece, principal piece and end piece of sperm flagella. Red arrows indicated missing central microtubules. Scale bar: 100 nm. M, mitochondrial sheath; MT, microtubules; ODF, outer dense fibers; CP, central microtubules; FS: fibrous sheath. *n* = 3 biologically independent samples were included in each group (**A**, **C**, **G**).

spermatocytes to sperm. During meiosis of spermatocytes, microtubules can form spindles to segregate homologous chromosomes in metaphase I and segregate replicated sister chromatids in metaphase II[53]. To analyze the effects of STYXL1 on spindle formation, we examined the metaphase spermatocytes in stage XII after deletion of *Styxl1*. The results showed that the spindles were normally formed with chromosomes aligned in the equatorial plane in *Styxl1⁻/⁻* metaphase spermatocytes (Supplementary Fig. 5C), indicating that *Styxl1* deletion did not affect the formation of spindles of spermatocytes. Thus, STYXL1 is important for the tubulin polymerization and microtubule assembly in sperm axoneme but not for spindle formation during meiosis.

## Discussion

Sperm axoneme in flagella is assembled by tubulin polymerization, it is not known how this process is regulated. Here, we identified a germ cell-specific pseudophosphatase, STYXL1, which interacted with chaperone CCT complex. STYXL1 regulated the assembly of chaperone CCT complex and tubulin polymerization in sperm. Deletion of *Styxl1* led to male infertility and MMAF with defective axonemal microtubule arrangements.

Our data revealed the vital role of pseudophosphatase STYXL1 in the development of sperm flagella. Pseudophosphatase is a protein that is totally devoid of catalytic activity, an estimated 13.8% of phosphatome are defined as pseudophosphatases[54]. Pseudophosphatases are increasingly viewed as signaling pathways regulators, such as competing with the active enzyme for substrate binding, serving as signaling integrators and modulators, or restricting the interactor to a subcellular localization in cellular processes[54–56]. Pseudophosphatase misregulations have been found to be related to various diseases such as leukemia, breast cancer, obesity and Charcot-Marie-Tooth disorder[54,57,58]. However, the role of pseudophosphatases in spermatogenesis and male sterility is rarely studied. Pseudophosphatase STYX complexes with CRHSP-24, a testicular RNA-binding protein, and is essential for normal spermiogenesis. Ablation of *Styx* leads to sperm head malformations but normal sperm flagella formation in mouse[59], and the detailed molecular mechanism remains unknown. We found that deletion of *Styxl1* resulted in defects in both sperm head and flagella, indicating different functions between STYXL1 and STYX. Moreover, our findings revealed that pseudophosphatase STYXL1 interacted with CCT complex subunits by its non-phosphatase region, regulated the assembly of CCT complex and tubulin polymerization in sperm flagella, providing evidence for STYXL1's role in sperm flagella development.

We found that chaperone CCT complex folds tubulin during tubulin polymerization in sperm, which is regulated by a germ cell-specific co-factor STYXL1. Sperm formation involves complex protein folding processes. For example, we recently found that male germ cell-specific RibosomeST predominantly regulates the folding of a subset of male germ cell-specific proteins in sperm formation[60]. Deletion of RibosomeST in mice results in increased ratios of abnormal sperm head and tail with microtubule defects in axoneme, similar to the phenotypes observed in *Styxl1⁻/⁻* sperm. Newly translated proteins by ribosomes are usually subjected to further folding by the chaperone.

Chaperone CCT complex provides chemical and topological directives that shape the folding landscape of its substrates, tubulin[15]. This tubulin folding pathway is guided by site-specific interactions with conserved regions in the CCT complex chamber. Here, we found that STYXL1 acted as a co-factor of CCT complex, and regulated the assembly of CCT complex in the tubulin folding/polymerization pathway. Deletion of *Styxl1* generated disorganized microtubules in sperm flagella. However, the spindle microtubules were normal in *Styxl1⁻/⁻* spermatocytes. Thus, STYXL1 regulated CCT complex-mediated tubulin polymerization process in sperm but not in spindle formation in spermatocytes.

Our results showed that the function of CCT Complex is regulated by germ cell-specific STYXL1 and is essential for sperm flagella development. In eukaryotic life, the chaperonin CCT Complex is indispensable for folding the cytoskeletal proteins actin and tubulin along with an estimated 10% of the remaining proteome[61]. Although CCT complex has a certain structure with eight subunits, it is obvious that the functions of the CCT complex are varied and related with lots of diseases. For example, CCT2 mutations can cause Leber Congenital Amaurosis[62] and mutation in CCT5 causes autosomal recessive mutilating sensory neuropathy[63]. As shown in the previously published papers, *Pdcl2⁻/⁻* mice displayed malformed and immotile sperm[64]. PDCL2 can interact with CCT and actin, based on which the authors speculated that PDCL2 might regulate actin production by CCT. However, actin folding was not studied and the function of CCT was not evaluated either. Deletion of *Tulp2* generated defective sperm tail structures and TULP2 interacted with CCT8[65]. The authors speculated that TULP2 might be the substrate of CCT8 and be correctly folded by the CCT complex, however, they didn't provide experimental evidence to confirm TULP2 as a substrate of CCT complex. *Cct6b⁻/⁻* mice showed normal spermatogenesis with normal sperm motility and male fertility[66]. Mild increase in the ratio of sperm nuclear base bending was observed. These data indicated that the testis-specific protein CCT6B is not essential for murine spermatogenesis. It is possible that CCT6B is not important for CCT complex assembly or spermatogenesis, and CCT6A might play major regulatory roles in spermatogenesis. Thus, the regulation of CCT complex in sperm flagella formation still remains not well known. Here, in our study, we found STYXL1 as a crucial CCT complex regulator. STYXL1 is essential for the formation of the intact CCT complex during spermiogenesis, working together to fold tubulin and facilitate microtubule polymerization in sperm axonemal formation. However, how STYXL1 regulates the assembly of CCT complex remains to be further studied.

In conclusion, germ cell-specific STYXL1 is essential in the regulation of sperm flagella formation, by interacting with CCT complex subunits and promoting CCT complex assembly in sperm flagellar microtubule polymerization.

## Methods

### All antibodies are given in Supplementary Table 1

**Mice.** C57BL/6J mice were housed in a specific-pathogen-free animal facility, provided with standard food, given free access to hypochlorous

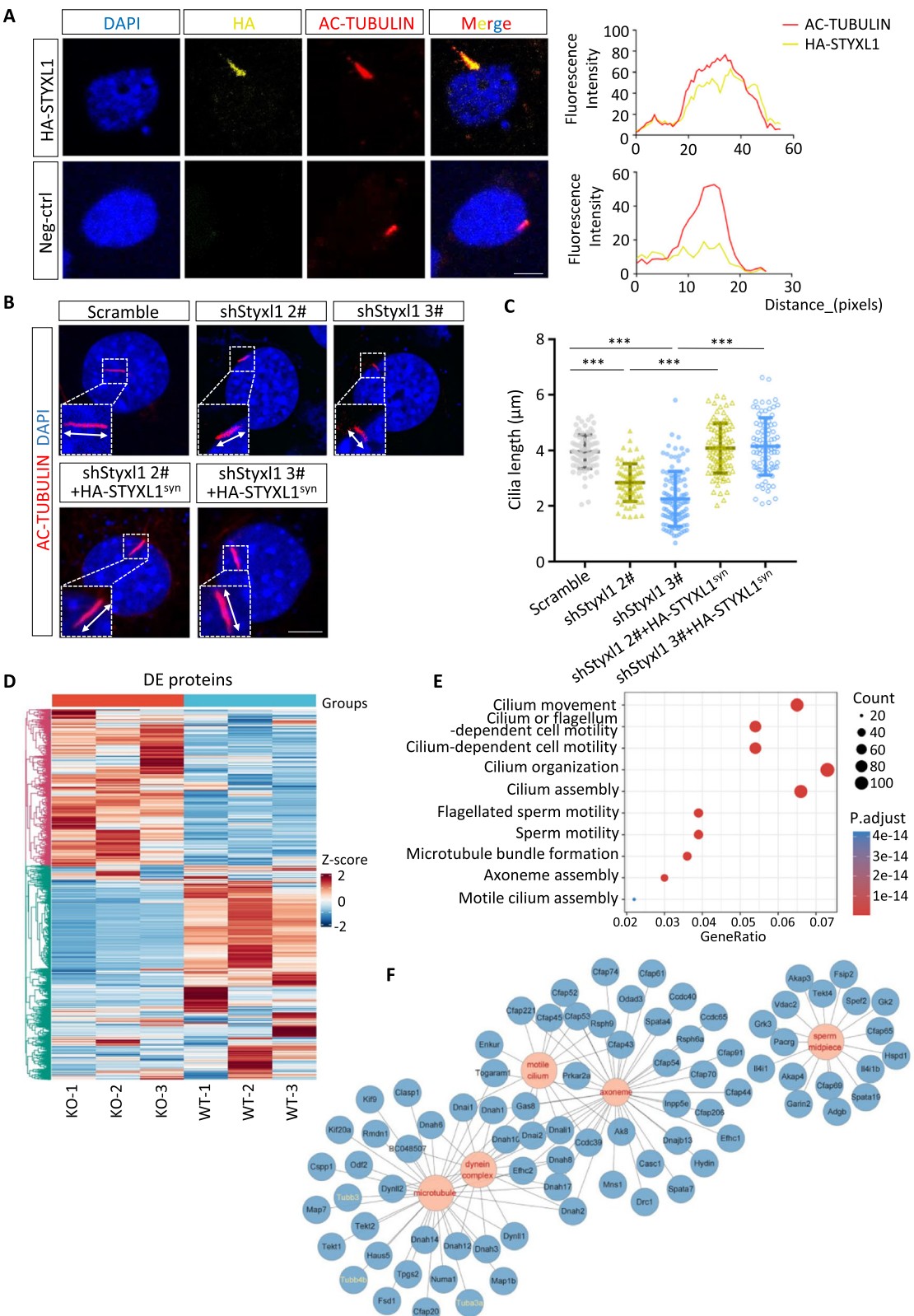

**Fig. 4 | Dysregulation of flagellar proteins in *Styxl1⁻ᐟ⁻* sperm. A** Colocalization analysis of HA-STYXL1 (yellow) and AC-TUBULIN (red) in primary cilia of HA-STYXL1 overexpressed NIH3T3 cells after serum starvation. Scale bar: 5μm.
**B**, **C** Immunofluorescence analysis (**B**) and length statistics (**C**) of primary cilia in NIH3T3 cells treated by *Styxl1* shRNA with or without HA-STYXL1^syn overexpression after serum starvation (*n* = 87 cells examined for Scramble group, *n* = 73 cells examined for shStyxl1 2# group, *n* = 106 cells examined for shStyxl1 3# group, *n* = 92 cells examined for shStyxl1 2# + HA-STYXL1^syn group, *n* = 86 cells examined

for shStyxl1 3# + HA-STYXL1^syn group over three independent experiments).
****p* < 0.0001 using one-way ANOVA followed by Dunnett's multiple comparisons test. AC-TUBULIN (red) and DAPI (blue) were used to stain cilia and nuclei, respectively. Scale bar: 5μm. **D** Heatmap of differentially expressed proteins between *Styxl1⁺ᐟ⁺* and *Styxl1⁻ᐟ⁻* sperm. Enriched terms of biological processes (**E**) and cellular components (**F**) are shown. ***, *p* < 0.001. Data are presented with the mean ± SD. Source data are provided as a Source Data file.

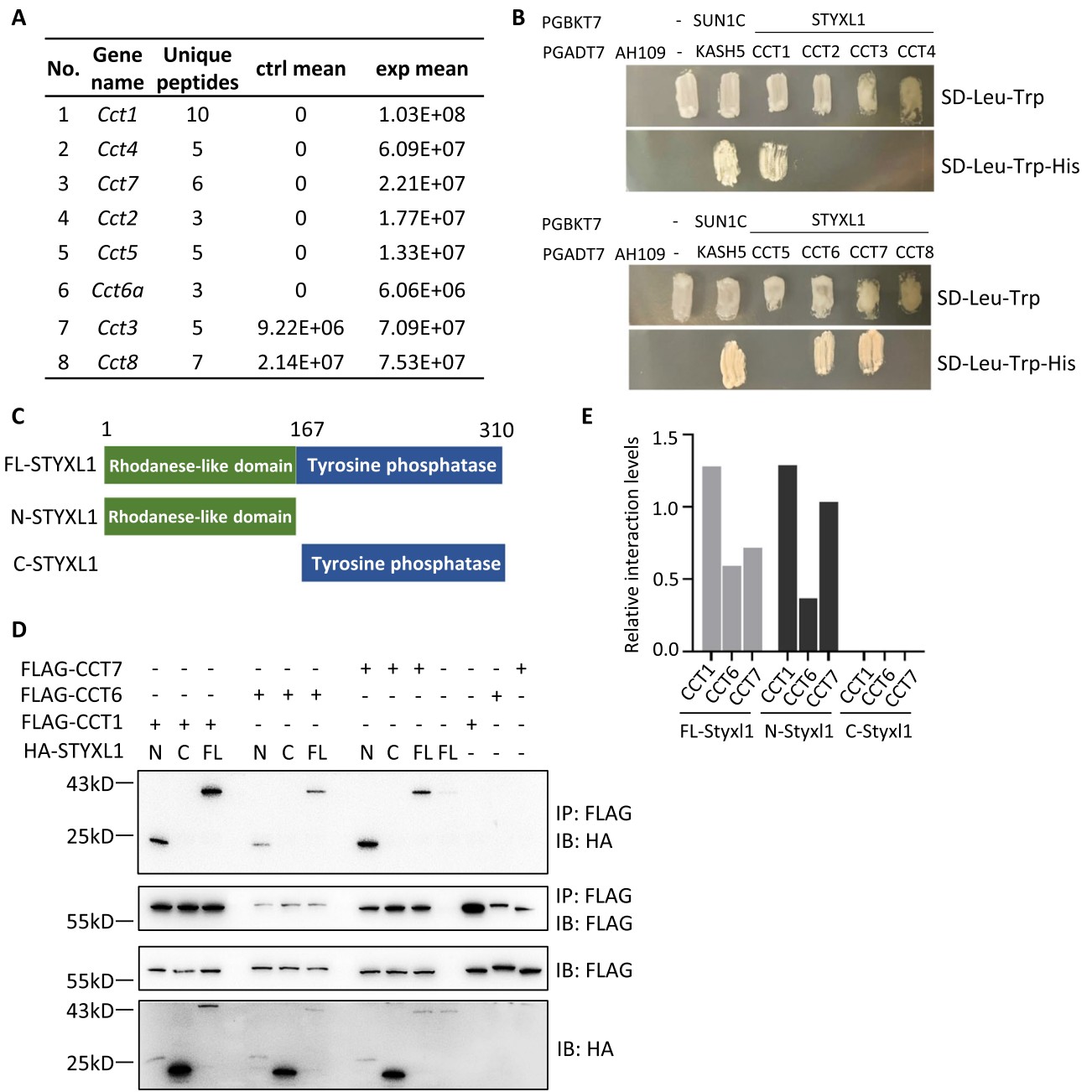

**Fig. 5 | The interactions of CCT complex subunits with STYXL1. A** The list of CCT complex subunits identified by MS in GST pull-down assay of STYXL1. **B** Yeast two-hybrid assay (Y2H) for validating interactions between STYXL1 and CCT complex subunits. SUN1C and KASH5 were used as positive controls, and empty vectors of PGBKT7 (-) and PBAKT7 (-) were used as negative controls. **C** Schematic representation of full-length (FL), N-terminal (N, 1-167 aa), C-terminal (C, 168-310 aa) of STYXL1. **D** co-IP assay of interactions between full length, N-terminal or C-terminal regions of STYXL1 and CCT1, CCT6 or CCT7. **E** The relative interaction levels between different regions of STYXL1 and CCT1, CCT6 or CCT7 normalized by input levels. $n$ = 3 biologically independent samples were included in each group (**A**, **B**).

weak-acid water, and maintained under a 12 h–12 h light–dark cycle with the temperature 24 ± 1 °C and humidity at 50 ± 10%. The animal study protocols were approved by the Institutional Animal Care and Use Committees of Nanjing Medical University (IACUC-1810007-2). All of the animal experiments were approved by the Institutional Animal Care and Use Committee of Nanjing Medical University and performed according to the guidelines and regulations of the Committee. CRISPR/Cas9 technology was used to generate the *Styxl1* knockout mice. Briefly, the single guide RNAs (sgRNAs) were designed to target exon 5 of *Styxl1*, Cas9 mRNA, and sgRNA were transcribed in vitro and then injected into the cytoplasm of fertilized zygotes. The injected zygotes developed to

blastocysts in vitro before transferring into the pseudopregnant C57BL/6J females to obtain the founders. Primers used to genotype: forward primer: 5'-GGTGTCTGCCTGTAGAACTCCTA-3', reverse primer: 5'-TGTTGTAACTCCAGCCTGTGTCA-3'.

**Cell culture and transfections.** The HEK293T (ATCC, CRL-3216) and NIH3T3 cells (ATCC, CRL-1658) were used in this study. Cells were cultured in DMEM with 10% FBS, penicillin (100 U/ml) and streptomycin (100 g/ml) in 37 °C and 5% $CO_2$. HEK293T and NIH3T3 lines were authenticated using STR profiling test by Shanghai Biowing Applied Biotechnology Co., Ltd and tested negative for mycoplasma

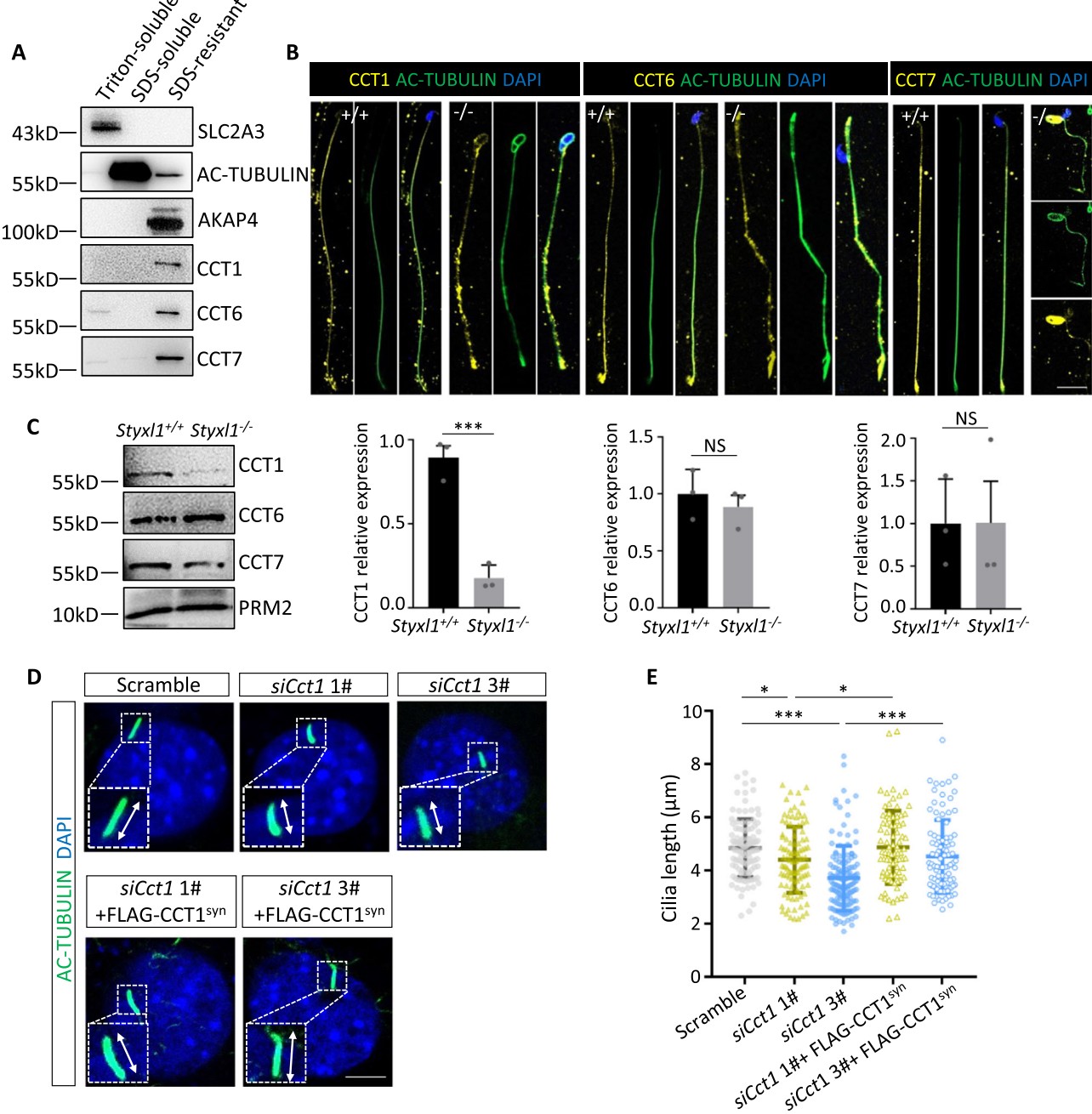

**Fig. 6 | The expression and function of CCT complex subunits. A** Western blot analysis of CCT1, CCT6 and CCT7 in Triton-soluble, SDS-soluble and SDS-resistant fractions of mouse sperm. **B** Immunofluorescence analysis of AC-TUBULIN (green) and CCT1, CCT6 or CCT7 (yellow) in *Styxl1*$^{+/+}$ and *Styxl1*$^{-/-}$ sperm with nuclei stained by DAPI (blue). Scale bar: 10μm. **C** Western blot and quantitative analysis of CCT1, CCT6 and CCT7 in sperm lysate from *Styxl1*$^{+/+}$ and *Styxl1*$^{-/-}$ mice (*n* = 3 mice per group). PRM2 was used as a loading control. ***$p$ = 0.0010 for CCT1, NS $p$ = 0.5249 for CCT6, NS $p$ = 0.9908 for CCT7 using two-tailed Student's t-test. **D, E** Primary cilia stained with AC-TUBULIN (red) and DAPI (blue) after serum starvation in NIH3T3 cells treated with *Cct1* siRNAs with or without FLAG-CCT1$^{syn}$ overexpression

(**D**) and quantitative analysis of cilia length (**E**) (*n* = 59 cells examined for Scramble group, *n* = 57 cells examined for *siCct1 1#* group, *n* = 86 cells examined for *siCct1 3#* group, *n* = 37 cells examined for *siCct1 1#* + FLAG-CCT1$^{syn}$ group, *n* = 48 cells examined for *siCct1 2#* + FLAG-CCT1$^{syn}$ group over three independent experiments). *$p$ = 0.0400 for Scramble vs *siCct1 1#*, *$p$ = 0.0382 for *siCct1 1#* vs *siCct1 1#* + FLAG-CCT1$^{syn}$, ***$p$ < 0.0001 for Scramble vs *siCct1 3#*, ***$p$ < 0.0001 for *siCct1 3#* vs *siCct1 3#* + FLAG-CCT1$^{syn}$ using one-way ANOVA followed by Dunnett's multiple comparisons test. Scale bar: 5 μm. NS, not significant; *, $p$ < 0.05; **, $p$ < 0.01; ***, $p$ < 0.001. Data are presented as mean ± SD. *n* = 3 biologically independent samples were included in each group (A, B). Source data are provided as a Source Data file.

contamination using TransDetect PCR Mycoplasma detection kit (Transgen, Beijing, China). Transfections were performed with ExFect Transfection Reagent (Vazyme) according to the manufacturer's instructions.

**RNA interference.** shRNAs were obtained from Pharmacorelab and target sequences are included: *Styxl1* shRNA (2#):

5'-GCTGTGGAATTTGGACAAA-3', *Styxl1* shRNA (3#): 5'-GCCA-TACCCAGTTGAGATA-3'. siRNA oligonucleotides were synthesized from GenePharma. The sequences of siRNAs are as follows: *Cct1* siRNA (1#): 5'-CCACCAUCCUGAAGUUACUTT-3', *Cct1* siRNA (3#): 5'-GGAGCGCUCUUUACAUGAUTT-3'. Transfection of siRNAs using siRNA-Mate (GenePharma) was performed according to the manufacturer's instructions.

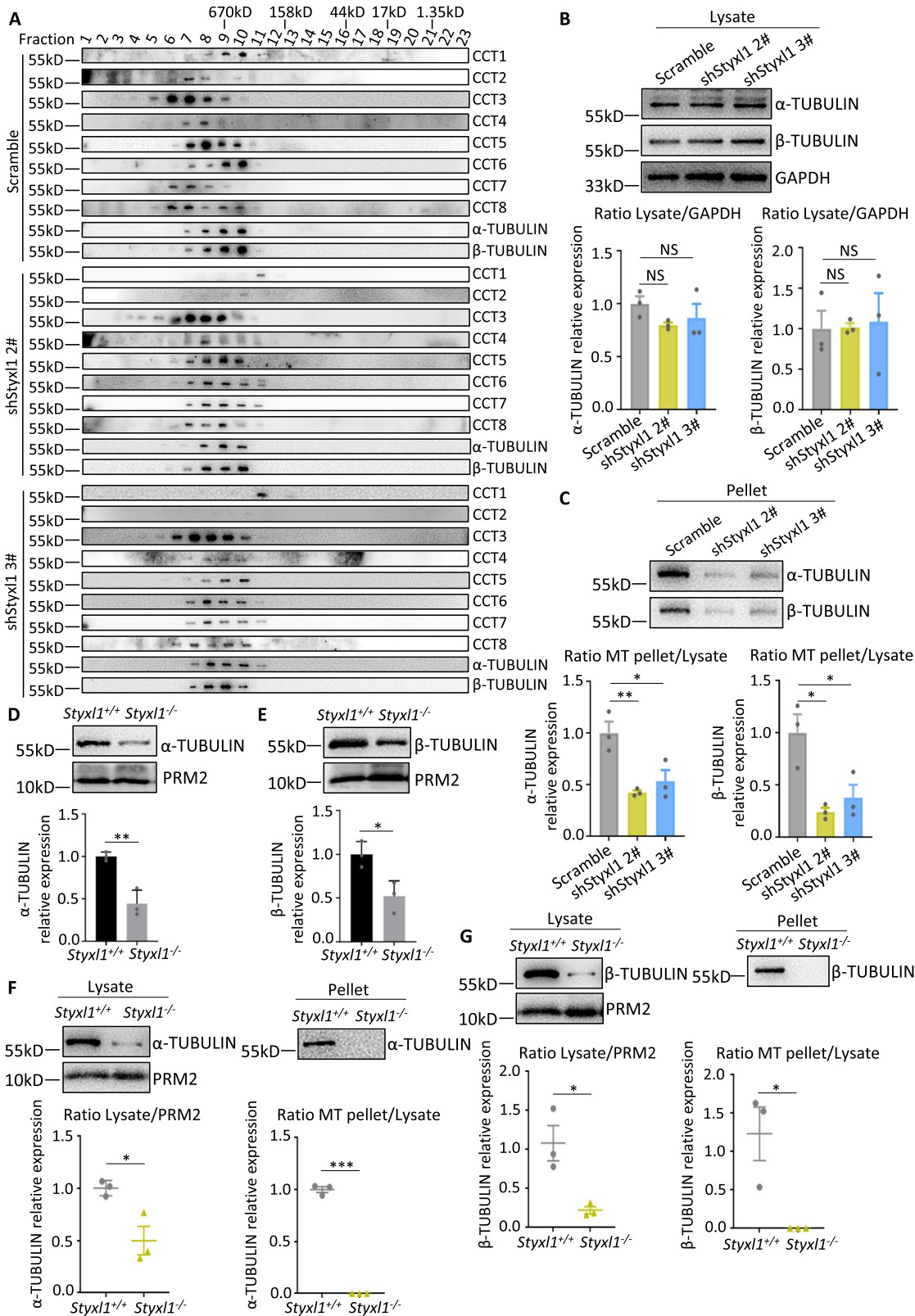

**RNA isolation and quantitative real-time PCR (qRT-PCR).** Total RNA was extracted with RNAiso Plus (Takara) and reverse transcribed into cDNA with PrimeScriptRT Master Mix (Takara). cDNA was then subsequently used for quantitative RT-PCR by Genious 2X SYBR Green Fast qPCR Mix (Abclonal) on a QuantStudio 7 (Applied Biosystems) system according to the manufacturer's instruction. The comparative ΔΔCt (cycle threshold) method that can calculate the changes in relative gene expression levels was used to analyze qRT-PCR data. Primers used in qRT-PCR are provided in the Supplementary Table 2.

**Sperm analysis.** Sperms were squeezed out from the cauda epididymis and incubated in mHTF media (Irvine Scientific) supplemented

**Fig. 7 | Abnormalities of CCT complex assembly and tubulin polymerization after *Styxl1* deletion. A** Gel filtration analysis of CCT complex proteins and CCT substrate proteins α-TUBULIN and β-TUBULIN in scramble or *Stxyl1* shRNA treated NIH3T3 cells. *n* = 3 biologically independent samples were included. **(B, C)** Western blots of α-TUBULIN and β-TUBULIN in the total lysate (α-TUBULIN: NS *p* = 0.2634 for Scramble vs shStyxl1 2#, NS *p* = 0.4886 for Scramble vs shStyxl1 3#; β-TUBULIN: NS *p* = 0.9977 for Scramble vs shStyxl1 2#, NS *p* = 0.9546 for Scramble vs shStyxl1 3# using one-way ANOVA followed by Dunnett's multiple comparisons test) **B** and pellet fractions (α-TUBULIN: **\*\****p* = 0.0070 for Scramble vs shStyxl1 2#, \**p* = 0.0182 for Scramble vs shStyxl1 3#; β-TUBULIN: \**p* = 0.0101 for Scramble vs shStyxl1 2#, \**p* = 0.0239 for Scramble vs shStyxl1 3# using one-way ANOVA followed by Dunnett's multiple comparisons test) **C** of scramble or *Stxyl1* shRNA treated NIH3T3

cells in the MT sedimentation assay, and quantification of tubulin levels in corresponding fractions (*n* = 3 biological replicates per group). Data were presented with the mean ± SEM. **D, E** Western blots of α-TUBULIN and β-TUBULIN in WT and *StyxI1$^{-/-}$* sperm with PRM2 as a loading control (*n* = 3 mice per group). **\*\****p* = 0.0047 for α-TUBULIN, \**p* = 0.0229 for β-TUBULIN) using two-tailed Student's t-test. Data are presented with the mean ± SD. **F, G** Western blot and quantification analysis of α-TUBULIN (**F**) (\**p* = 0.0240 and **\*\*\****p* < 0.0001 using two-tailed Student's t-test) and β-TUBULIN (**G**) (\**p* = 0.0205 for ratio Lysate/PRM2, \**p* = 0.0241 for ratio MT pellet/Lysate using two-tailed Student's t-test) in the total lysate and pellet in the MT sedimentation assay of *StyxI1$^{+/+}$* and *StyxI1$^{-/-}$* sperm (*n* = 3 mice per group). Data were presented with the mean ± SEM. NS, not significant, \*, *p* < 0.05, \*\*, *p* < 0.01, \*\*\*, *p* < 0.001. Source data are provided as a Source Data file.

with 10% FBS at 37 °C for 10 min. Diluted sperm were counted by a hemocytometer, and analyzed for sperm motility assay using Hamilton Thorne's CASA system.

**Immunoblotting.** Protein extracts were lysed in RIPA lysis buffer (Beyotime) with 1× protease inhibitor cocktail (Biomake), subjected to SDS-PAGE, and then transferred onto PVDF membranes. The membranes were blocked with 5% non-fat milk at room temperature for 2 h, and incubated with corresponding primary antibodies and secondary antibodies. After being washed with 0.1% TBS-Tween-20 (TBST) for three times, the protein bands were detected by the High-sig ECL western blotting substrate (Tanon).

**Fractionation of sperm.** The sperm was fractionated as previously described[67]. Sperm from cauda epididymis were collected in 1% Triton X-100 lysis buffer (50 mM NaCl, 20 mM Tris-HCl, pH 7.5) at 4 °C for 2 h and centrifuged at 15,000 *g* for 10 min to obtain the Triton-soluble fraction (supernatant). The pellet was then resuspended in 1% SDS lysis buffer (75 mM NaCl, 24 mM EDTA, pH 6.0) for 1 h and centrifuged at 15,000 *g* for 10 min to obtain the SDS-soluble fraction (supernatant) and SDS-resistant fraction (pellet). Each fraction was then subjected to immunoblotting for further analysis.

**Histological analysis.** Samples were fixed in modified Davidson's Fluid (MDF) (30% formaldehyde, 15% ethanol, 5% glacial acetic acid, and 50% distilled H$_2$O), dehydrated with gradient ethanol, embedded in paraffin and sections were then cut at a 5-μm thick for further histological examination. Sections were then deparaffinized, rehydrated, and then stained with periodic acid-Schiff's (PAS) reagent for PAS staining or stained routinely with Hematoxylin and Eosin (H&E) for H&E staining.

**Transmission electron microscopy.** For transmission electron microscopy, samples were prepared by fixation in 2.5% glutaraldehyde, dehydration in ethanol, embedding in Epon812 (Sigma) and polymerization. Ultrathin sections were cut and stained with uranyl acetate and lead citrate and observed by transmission electron microscope (JEOL, JEM-1010).

**Immunofluorescence (IF).** For paraffin sections, the IF assays were performed as described previously[68]. For IF staining of sperm, sperm spread onto microscope slides were air-dried and fixed with 4% paraformaldehyde for 40 min. Following PBS washing, the slides were then blocked with 5% BSA for 2 h and incubated with primary antibodies. For IF of cultured cells, NIH3T3 cells were plated on glass coverslips and cultured in serum starvation for 48 h. The cells were then fixed with 4% paraformaldehyde, washed with PBS, blocked with 5% BSA and stained with the primary and secondary antibodies. For IF staining of testicular suspension, the testes were digested into single cell suspension using collagenase and trypsin, fixed with 4% paraformaldehyde and spread onto microscope slides for further staining. The IF images were acquired with an LSM 800 microscope (Zeiss, Germany) or SP8 microscope (Leica, Germany).

**GST pull-down assay.** The mouse *Styxl1* cDNA fused with a GST-MBP tag or GST-MBP tag only was cloned into a pTXB vector and expressed in *E. coli*. GST-MBP–STYXL1 protein was incubated with glutathione–Sepharose 4B (GE Healthcare) at 4 °C overnight with GST-MBP protein as a control. WT testes lysate were extracted with RIPA lysis buffer (Beyotime) and then pre-cleared by incubating with glutathione–Sepharose 4B. The pre-cleared testes lysate was then incubated with GST-MBP–STYXL1 or GST-MBP protein immobilized glutathione–Sepharose 4B at 4 °C for 4-6 h and the beads were washed with wash buffer (50 mM Tris-HCl, pH 8.0, 150 mM NaCl, 2 mM MgCl$_2$ and 5% glycerol) for three times. Then the beads were eluted with 20 mM GSH and subjected to SDS-PAGE for Coomassie blue staining. The whole bands from the Coomassie blue-stained gel were subjected to LC-MS/MS analysis. Any protein with more than two unique peptides identified in the GST pull-down lane with foldchange greater than 3 were considered for analysis.

**Yeast two-hybrid assay.** Mouse Styxl1 was cloned into pGBKT7 vector as bait and mouse CCT1-CCT8 were cloned into pGADT7 as prey. The bait and prey plasmids were co-transformed into the yeast AH109 competent cells, which were cultured at 30 °C for 2–3 days and selected on SD-Leu-Trp-His plates.

**Co-immunoprecipitation.** Full length or truncations of STYXL1 were co-transfected with CCT1, CCT6 or CCT7 into HEK293T cells for 48 h. The cells were then lysed in iced IP lysis buffer (Thermo Fisher Scientific) supplemented with 1× protease inhibitor cocktail (Biomake), and centrifuged at 12, 000 *g* for 20 min at 4 °C. The cell lysates were incubated with anti-DYKDDDDK Resin (GenScript) at 4 °C overnight, and washed with wash buffer (20 mM Tris, pH 7.4, 150 mM NaCl, 0.5% Triton X-100, 1 mM EDTA) for three times. The resins were eluted with sample buffer containing 1% SDS at 95 °C for 10 min, and then analyzed by immunoblotting.

**Gel filtration assay.** Cell extract was chromatographed over a Superose-6 10/300 GL column (GE Healthcare) and collected 1 mL per fraction. The fractions were then detected by immunoblotting. The Gel Filtration Standard (Bio-Rad) containing a mixture of 1.35 to 670 kDa molecular weight markers was used to calibrate the column.

**Microtubule (MT) sedimentation assay.** Microtubule (MT) sedimentation assay was performed as previously described[69]. Briefly, proteins were extracted in MT buffer (100 mM PIPES, pH 6.9, 1 mM MgCl$_2$ and 1 mM EGTA) with protease inhibitor (Biomake) and incubated on ice for 15 min. The lysate was centrifuged at 256,302 *g* at 4 °C for 30 min by ultracentrifugation. For microtubule polymerization, 20 μM taxol (Sango Biotech) and 1 mM GTP (Beyotime) were added to the lysate. After incubation at 37 °C for 30 min, the lysate was loaded onto a sucrose cushion supplemented with 1 M sucrose, 10 μM taxol and 0.5 mM GTP. Followed by centrifuging at 256,302 *g* at 37 °C for 30 min, the pellet was re-suspended in protein loading buffer and

boiled at 95 °C for 10 min for further analysis. Ultracentrifugation was performed in a Beckman SW 60 Ti rotor in Optima XPN-100 ultra-centrifuge (Beckman).

**Protein quantification by PRM.** PRM was performed as previously described[60,70]. Peptide samples digested from the testes by trypsin were combined with stable isotope-labelled peptides (Supplementary Table 3) and injected into the easy-nLC 1200 HPLC system (Thermo Fisher Scientific). Quantification was performed using a scheduled method on an LTQ Fusion Lumos mass spectrometer (Thermo Fisher Scientific). Skyline Daily software was used to process the PRM data. For relative quantification, three transitions per precursor were used to quantify the targeted peptides in the samples, peptide intensity was calculated as the sum of the corresponding transition peak areas, and the protein expression was measured based on the intensity of light peptides normalized against the heavy peptides.

**Protein preparation and TMT labelling.** Epididymal sperm were lysed in protein extraction buffer (8 M Urea, 75 mM NaCl, 50 mM Tris-HCl, pH 8.2, 1% EDTA-free protease inhibitor, 1 mM NaF, 1 mM β-glycerophosphate, 1 mM sodium orthovanadate, 10 mM sodium pyrophosphate), followed by reduction, digestion, and desalting. For TMT labelling, purified peptides were reconstituted in 200 mM triethylammonium bicarbonate (TEAB) and labelled with TMT-6plex reagent (Thermo Fisher Scientific) to react for 1 h at room temperature according to the manufacturer's instructions. The reaction was then quenched by 5% hydroxylamine for 15 min. After TMT labeling, all six samples were combined, purified by an OASIS HLB 1-cc Vac cartridge (Waters), and then lyophilized for subsequent analysis.

**High-pH reverse phase fractionation.** High-pH reverse phase fractionation was performed as described[60]. The mixture peptides were fractionated with an Xbridge™ BEH130 C18 column (300 μm × 150 mm, 1.7 μm; Waters) using the ACQUITY UPLC M-Class system (Waters). A 128 min gradient (3% buffer B for 14 min, 3–8%B for 1 min, 8–29%B for 71 min, 29–41% B for 12 min, 41–100% B for 1 min, 100% buffer B for 8 min, 100–3% B for 1 min, followed by 20 min at 3% B) was employed with buffer A (20 mM ammonium formate, pH 10) and buffer B (100% ACN). A total of 30 fractions were collected using a non-adjacent pooling scheme. The peptides were dried with a SpeedVac concentrator (Labconco).

**Mass spectrometry analysis.** Peptides were resuspended in 0.1% FA and analyzed with an analytical column (75 μm × 150 mm, 1.9 μm, Dr. Maisch) using a 95 min linear gradient (3% to 5% buffer B for 5 sec, 5% to 15% buffer B for 40 min, 15% to 28% buffer B for 34 min and 50 sec, 28% to 38% buffer B for 12 min, 38% to 100% buffer B for 5 sec, 100% buffer B for 8 min) at 300 nl/min. The Orbitrap Fusion Lumos was configured with a resolution of 60 K for MS1 and 15 K for MS/MS, and the data were acquired by Xcalibur 4.3.73.11 (Thermo Fisher Scientific).

All raw files were searched with MaxQuant software (Version 2.2.0.0) using mouse protein sequences obtained from the Universal Protein Resource database (UniProt, release 2022_01). Both the protein and peptide FDR cut-off was set to 0.01. Full cleavage by trypsin, and the number of maximal miscleavage sites was permitted to two. Carbamidomethylation (C) on cysteine and TMT reagent adducts on lysine and peptide amino termini was considered as fixed modifications. Oxidation (M) and acetylation (protein N-term) were set as variable modifications. For TMT experiment settings, reporter ion MS2 type and 6plex TMT isobaric labels were selected. The corrected TMT reporter intensities were used for TMT-based protein quantification. Statistical significance was determined using the unpaired two-tailed Student's t-test. A protein with $p$ value < 0.05 and foldchange greater than 2 was considered significantly differentially expressed.

**Statistical analysis.** Two-tailed Student's t-test or one-way ANOVA followed by Dunnett's multiple comparisons was used to determine the statistical significance. Each experiment was performed at least three times, and $p < 0.05$ was considered significant.

**Reporting summary**
Further information on research design is available in the Nature Portfolio Reporting Summary linked to this article.

## Data availability
The MS proteomics data generated in this study have been deposited in the PRIDE database under accession code PXD046203. Source data are provided with this paper.

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

## Acknowledgements

This work was supported by the National Key R&D Program (2021YFC2700200 to X.G.), the National Natural Science Foundation of China (82221005 to X.G., 82001611 to Y.L., 81971439 to X.G., 82201764 to T.Z.), the fund from Health Commission of Jiangsu Province (M2020071 to Y.L.), Natural Science Foundation of Jiangsu Province (BK20220316 to T.Z.), China Postdoctoral Science Foundation (2022M711676 to T.Z.), and Natural Science Foundation of the Jiangsu Higher Education Institutions (22KJB310011 to C.S).

## Author contributions

Y.Ch., M.L., H.T., and Y.Q. carried out the main part of the experiment. M.G. performed the histology experiments. Y.G. and X.Zha. contributed to the proteomics analysis. Y.Cu., X.Zho., T.Z., and H.Z. performed animal work. X.G., Y.L., C.S., and Y.Ch. wrote and revised the manuscript. X.G., Y.L., and C.S. conceived, designed, and supervised the project. All authors reviewed the manuscript and approved it for submission.

## Competing interests

The authors declare no competing interests.
