## [Peer Review File · Nature Communications]

STYXL1 regulates CCT complex assembly and flagellar tubulin folding in sperm formationREVIEWER COMMENTS

Reviewer #1 (Remarks to the Author):

Chen et al report on the role of Styxl1 in sperm formation. They identify this pseudo-phosphatase as one of several phosphatases that may play role in spermiogenesis but remains uncharacterized. They then generate a KO mouse model and show that loss of Styxl1 results in male infertility arising from severe defects in sperm formation. The spermatids notably display abnormal microtubule formation in the flagellum. Proteomics and imaging demonstrate that this protein is associated with ciliary proteins. Mechanistically, the N-term domain of Styxl1 binds to CCT proteins, which form a chaperone complex involved in microtubule assembly. Loss of Styxl1 impairs CCT complex formation, suggesting that Styxl1's role is in the proper function of CCT.

Overall, this is a very good article. The text is generally well written (except for a few mistakes/typos), and the data is presented in a very clear and logical manner. The findings appear quite robust and I have no concerns regarding the phenotypic characterization of the Styxl1-KO mouse strain and the effect on sperm formation. This work is significant for the field of spermatogenesis and axoneme/cilia formation in general. However, the interpretation of the structure and the data regarding the CCT:Styxl1 interaction of Styxl1 is questionable, and the mechanism of action of Styxl1 has some gaps that could be investigated further. Here are my main comments/suggestions to improve the manuscript:

- 1) It appears that the authors have not realized that the N-terminal segment of Styxl1 (1-167) is a rhodanese-like domain, which is found notably in dual-specificity MAPK-phosphatase such as MPK5 where they bind kinase domains to enable the phosphatase domain to engage with the kinase (see Zhang et al 2011 Science Signaling). This should be mentioned, analyzed by comparing their structures, and discussed appropriately.
- 2) Furthermore, the fact that the N-term of Styxl1 is a rhodanese-like domain implies that it cannot be cut in two parts as in Figure 5F without unfolding the rest of the domain, thus invalidating whatever binding data is obtained with such deletions. In addition, the data presented in Figure 5F does not support the author's conclusion. First of all, it is unclear why there is no inclusion of FL Styxl1 as a Flag-IP on the same blot. I understand this is presented in Figure 6E, but with immunoblots, it is not possible to compare the intensities across gels, and thus a FL control should have been included here as well. I would therefore remove Figure 5F altogether. Figure 5E remains valid, but the band intensities should be quantified.
- 3) It remains unclear how Styxl1 is acting on the CCT complex to modify its activity or assembly. To start with, it is hard to appreciate the extent of the molecular size changes induced by the loss of Styxl1 in Figure 7A. The authors should indicate where molecular weight standards run above the fraction numbers. Secondly, if CCT1 dissociates in absence of Styxl1, one would expect the whole CCT octamer to dissociate, yet other CCT subunits don't dissociate in the knock-down. This begs the question; how is the CCT reorganizing? Are other CCT subunits known to be able to replace CCT1? Please discuss in the context of what is known in the literature and provide further evidence for mechanism if available.
- 4) In the discussion, the authors write "Styxl1 is essential for the formation of the intact CCT complex". This is a very bold statement which should be qualified. The CCT is found throughout the body, whereas Styxl1 appears to be expressed only in testis; thus Styxl1 cannot be "essential" for CCT's assembly in general, but only in testis. This should be corrected.
- 5) In Suppl. Figure S5, the molecular weights of the bands don't make sense. GST is 26 kDa, thus it's unclear why the band appears at 55 kDa. Furthermore, the GST-Styxl1 band should be approximately 60 kDa, not 110 kDa. Furthermore, what are all those bands between 110 and 50 kDa? This should be revised and modified. Ideally, the authors should really show a gel with four lanes: GST alone, GST-Styxl1 alone, GST with lysate, GST-Styxl1 with lysate. And then, clearly indicate what the bands on the gel are.

Minor comments:

6) Line 92: "...Eya4 are essential for embryonic lethality". The authors probably meant to write: "...Eya4 are essential to prevent embryonic lethality"?

7) Figure 5C: "Tyrosine" not "Tryosine". You should also change the color scheme for the "helix/sheet-loop-helix/sheet" and replace with a single color for the N-term rhodanese-like domain.

8) Line 305: The study cited in reference 51 (Isrie et al) was later disproved by Hengel et al (2019; Eur J Med Genet). The mutation P311A is likely benign. Thus this mutation in Styxl1 does not cause neurological disorders nor epilepsy. Please correct.

Reviewer #2 (Remarks to the Author):

The authors describe the role of STYXL1 in the formation of sperm flagella. They show that deletion of Styxl1 in mice results in male infertility and microtubule defects of sperm flagella. Quantitative proteomic analysis of Styxl1^{-/-} sperm revealed mainly abnormal downregulation of flagella-related proteins, suggesting that STYXL1 impacts expression of axoneme proteins during development of sperm flagella. The authors also provide evidence, that STYXL1 interacts with the eukaryotic chaperonin CCT complex, likely regulating the folding and stability of sperm flagellar proteins during sperm flagellar formation.

We are just beginning to understand the causes of male infertility and especially pathomechanisms leading to multiple morphological abnormalities of sperm flagella (MMAF). Thus, the characterization of the role of the pseudophosphatase STYXL1 in sperm flagella formation and regulation of sperm motility is important and will add a novel gene when analysing genetic causes of male infertility and especially MMAF.

Overall, this manuscript adds knowledge on not only phenotype of Styxl1 mouse mutants, but also knowledge on the associated pathomechanisms related to male infertility.

However, three previously published papers already addressed regulation of or regulation by the chaperonin CCT complex during spermiogenesis using mouse models, without being referenced in this manuscript. Here, the authors overlooked the novelty of this study and findings in relation to these previously published studies, and therefore this study lacks a conceptual advance that is a standard in Nature publications:

Li M, Chen Y, Ou J, Huang J, Zhang X. PDCL2 is essential for spermiogenesis and male fertility in mice. *Cell Death Discov.* 2022 Oct 17;8(1):419. doi: 10.1038/s41420-022-01210-2. PMID: 36253364; PMCID: PMC9576706.

Zheng M, Chen X, Cui Y, Li W, Dai H, Yue Q, Zhang H, Zheng Y, Guo X, Zhu H. TULP2, a New RNA-Binding Protein, Is Required for Mouse Spermatid Differentiation and Male Fertility. *Front Cell Dev Biol.* 2021 Feb 18;9:623738. doi: 10.3389/fcell.2021.623738. PMID: 33763418; PMCID: PMC7982829.

Yang P, Tang W, Li H, Hua R, Yuan Y, Zhang Y, Zhu Y, Cui Y, Sha J. T-complex protein 1 subunit zeta-2 (CCT6B) deficiency induces murine teratospermia. *PeerJ.* 2021 Jun 1;9:e11545. doi: 10.7717/peerj.11545. PMID: 34141486; PMCID: PMC8176918.

Overall, the work supports the conclusions on STYXL1 function, but some major concerns and minor comments remain.

Major comments:

Figure 1 A and B: In the text and Figure 1 A, 23 testis-enriched phosphatases are mentioned, but in the heatmap (Figure 1B) only 22 are presented. What happened to number 23?

Figure 1 D: it is surprising that Styxl1 mRNA was only detectable in testis, using RT-PCR. According to publicly available data, STYXL1 shows similar expression in other tissues compared to testis. In addition, publicly available antibody stainings also demonstrate STYXL1 localization in diverse tissues,

not only testis. Please explain this discrepancy and/or provide additional data. Overall, the (q)RT-PCR results, also those presented in Figure S1, are not convincing.

Figure 2 A: The authors should provide evidence by Western blot analysis or immunofluorescence analysis, that STYXL1 is lacking in mutant mice. Several antibodies are commercially available that could have been used for this purpose and a full knock out mouse model should demonstrate complete absence of this protein.

Figure 3A and B: it is difficult to appreciate the differences between the control and mutant animals.

Figure 6 B, line 240: provided images do not convincingly demonstrate that loss of STYXL1 affects distribution of CCT1, CCT6, CCT7 in sperm flagella. I recommend to exchange these images showing STYXL1 deficient sperm with severe MMAF with images of less malformed sperm flagella. These should be available as suitable examples have been shown in Figure 2G and 3B (top). In addition, despite low sperm count compared to control (Figure 2B), only about 35-40% of those present with morphological abnormalities, therefore a more convincing IF analysis can and should be provided.

Line 121: it remains unclear, if the 4 bp and 74 bp deletion is on one the same or different alleles. If two independent mutant lines have been generated, which of those has undergone deeper analysis and if both have been analysed (and only results of one strain are presented) do they show the same results?

Line 127: The authors should provide videos from wildtype control and Styxl1^{-/-} sperm to provide additional evidence.

Line 174 following: please explain, why knockdown in NIH3T3 cells was used to assess the impact of STYXL1 on cilia in fibroblasts instead of taking fibroblasts from their established mouse line using standard protocols? Also, fibroblasts do not represent the right "test tube" to assess function of genes that are potentially involved in motile cilia generation or sperm cell maturation.

Line 350: Sperm motility defects in Styxl1 mutants are likely caused by MMAF. Here, the conclusion that STYXL1 is not only regulating sperm flagella formation but also sperm motility appears not appropriate, based on data provided.

Proteomics: The interaction data, especially regarding interaction with CCT1, CCT6 and CCT7 are convincing. However, regarding the LC-MS/MS approach, the threshold for differential expression (2fold) appears to be very low. Thus, the dataset is likely overestimated. A re-analysis using 3 to 5 fold difference as cutoff should be performed. Also, a great variability between samples is to be expected that could lead to false positive/false negative data due to the low cutoff. Thus, it is in addition important to know how many samples have been analyzed.

Minor comments:

Complete manuscript:

- please check and use correct capitalization of protein symbols.
- please check for spellings and grammar.

Reviewer #1 (Remarks to the Author):

Overall, this is a very good article. The text is generally well written (except for a few mistakes/typos), and the data is presented in a very clear and logical manner. The findings appear quite robust and I have no concerns regarding the phenotypic characterization of the Styx11-KO mouse strain and the effect on sperm formation. This work is significant for the field of spermatogenesis and axoneme/cilia formation in general. However, the interpretation of the structure and the data regarding the CCT:Styx11 interaction of Styx11 is questionable, and the mechanism of action of Styx11 has some gaps that could be investigated further. Here are my main comments/suggestions to improve the manuscript:

1) It appears that the authors have not realized that the N-terminal segment of Styx11 (1-167) is a rhodanese-like domain, which is found notably in dual-specificity MAPK-phosphatase such as MPK5 where they bind kinase domains to enable the phosphatase domain to engage with the kinase (see Zhang et al 2011 *Science Signaling*). This should be mentioned, analyzed by comparing their structures, and discussed appropriately.

Response: We appreciate for the suggestion by the reviewer. STYXL1 belongs to the large family of dual-specificity protein phosphatases (DUSPs) and possesses a rhodanese-like domain in N-terminal segment and a C-terminal tyrosine-protein phosphatase domain. In MAPK phosphatases (MKPs), also members of DUSPs, the N-terminal rhodanese-like domain can bind mitogen-activated protein kinase (MAPK) through its kinase interaction motifs (KIM) to downregulate the kinase activity and hence play an important role in controlling cellular responses and the physiological outcome of signaling (Zhang et al 2011 *Sci Signal*; PMID: 22375048). However, in STYXL1 protein the KIM motif of the rhodanese-like domain is mutated and lacks consecutive critical arginines (Hepworth EMW et al 2021 *Int J Mol Sci*; PMID: 34830476), which are required for MAPK docking (Qi et al 2022 *Sci Rep*; PMID: 35264672). Our study showed that MAPK was not identified in the proteomic analysis of co-immunoprecipitated proteins of STYXL1, supporting a different function of rhodanese-like domain of STYXL1 from that of MAPK kinase. Further yeast two-hybrid experiments and reciprocal co-immunoprecipitation analysis showed that the N-terminal rhodanese-like domain mediated the interactions between STYXL1 and CCT1, CCT6 and CCT7, which were independent of the tyrosine-protein phosphatase domain, indicating new functions of the rhodanese-like domain in DUSPs. Above analysis have been added in the revised manuscript (Page 10, line 244-257).

2) Furthermore, the fact that the N-term of Styx11 is a rhodanese-like domain implies that it cannot be cut in two parts as in Figure 5F without unfolding the rest of the domain, thus invalidating whatever binding data is obtained with such deletions. In addition, the data presented in Figure 5F does not support the author's conclusion. First of all, it is unclear why there is no inclusion of FL Styx11 as a Flag-IP on the same blot. I understand this is presented in Figure 6E, but with immunoblots, it is not possible to compare the intensities across gels, and thus a FL control should have been included here as well. I would therefore remove Figure 5F altogether. Figure 5E remains valid, but the band intensities should be quantified.

Response: Thanks for your constructive suggestions. Given that the N-terminal segment of

STYXL1 (1-167) is a rhodanese-like domain, it is not suitable to be cut into different parts. We have removed the original Fig. 5F in the revised manuscript. According to the reviewer's suggestion, we added quantification of band intensities, and calculated relative interaction levels by normalizing against the input level in **Fig. 5E** in the revised manuscript.

Figure 5E. The relative interaction levels between different regions of STYXL1 and CCT1, CCT6 or CCT7 normalized by input levels.

3) It remains unclear how Styxl1 is acting on the CCT complex to modify its activity or assembly. To start with, it is hard to appreciate the extent of the molecular size changes induced by the loss of Styxl1 in Figure 7A. The authors should indicate where molecular weight standards run above the fraction numbers.

Secondly, if CCT1 dissociates in absence of Styxl1, one would expect the whole CCT octamer to dissociate, yet other CCT subunits don't dissociate in the knock-down. This begs the question; how is the CCT reorganizing? Are other CCT subunits known to be able to replace CCT1? Please discuss in the context of what is known in the literature and provide further evidence for mechanism if available.

Response: We appreciate for these constructive suggestions. The Gel Filtration Standard (Bio-Rad) containing a mixture of markers with molecular weights from 1.35 to 670 kDa was used to calibrate the column. Elution volumes and approximate molecular weights of fractions were annotated at the top of the fraction numbers in **Fig. 7A**. During revision, we added Western blot results of CCT2, CCT4 and CCT5 proteins following gel filtration assay of wildtype and *Styxl1* knockdown NIH-3T3 cells. With the systemic analysis of all the 8 subunits of CCT octamer complex, we found that, in the elution fraction 8 of wildtype cells, all the 8 CCT complex subunits and α -TUBULIN and β -TUBULIN were co-eluted, indicating that tubulins are folded by intact 8-subunit CCT complex (**Fig. 7A**). However, in *Styxl1* knockdown NIH-3T3 cells, CCT1 was not co-eluted α -TUBULIN and β -TUBULIN, and shifted to later fractions with smaller molecular weights. CCT2 and CCT4 also shifted to elution fractions of smaller molecular weights in *Styxl1* knockdown NIH-3T3 cells. These results indicated that the formation of CCT octamer complex containing all 8 subunits were compromised in *Styxl1* knockdown cells. Previous studies have shown that CCT proteins can form micro-complexes with less than 8 CCT subunits (Liou AK et al 1997 *EMBO J*; PMID: 9250675). It seems that STYXL1 is essential for the assembly of intact CCT octamer complex with all 8 subunits to

fold α -TUBULIN and β -TUBULIN. Above data and analysis have been added in the revised manuscript (Page 12, line 288-301).

Figure 7A. Gel filtration analysis of CCT complex proteins and CCT substrate proteins α -TUBULIN and β -TUBULIN in scramble or *Styx11* shRNA treated NIH-3T3 cells.

4) In the discussion, the authors write “Styx11 is essential for the formation of the intact CCT complex”. This is a very bold statement which should be qualified. The CCT is found throughout the body, whereas Styx11 appears to be expressed only in testis; thus Styx11 cannot be “essential” for CCT’s assembly in general, but only in testis. This should be corrected.

Response: Thank you for your positive comment. We have corrected “Styx11 is essential for the formation of the intact CCT complex.” (line 344 in initial manuscript) to “STYXL1 is essential for the formation of the intact CCT complex during spermiogenesis.” (Page 15, line 394-395) in the revised manuscript.

5) In Suppl. Figure S5, the molecular weights of the bands don’t make sense. GST is 26 kDa, thus it’s unclear why the band appears at 55 kDa. Furthermore, the GST-Styx11 band should be approximately 60 kDa, not 110 kDa. Furthermore, what are all those bands between 110 and 50 kDa? This should be revised and modified. Ideally, the authors should really show a gel with four lanes: GST alone, GST-Styx11 alone, GST with lysate, GST-Styx11 with lysate. And then, clearly indicate what the bands on the gel are.

Response: Thanks for the reviewer’s comment. We’re sorry for the ambiguous descriptions about the GST pull-down assay. When we firstly expressed the mouse STYXL1 protein *in vitro*

using pTXB1-6xHis-mus Styxl1 plasmid (**Fig. R1**), the results showed that STYXL1 protein existed in the inclusion bodies (**Fig. R2**) and displayed poor solubility when using the *E. coli* expression systems. To improve the protein solubility, the maltose-binding protein (MBP) (about 44 kDa), a secretion-enhancing tag (Reuten R et al 2016 *PLoS One*; PMID: 27029048), was added to STYXL1, resulting a plasmid profile for STYXL1 fused to the GST, MBP and His tags shown below (**Fig. R3**). GST-MBP-His tag was only used as a control. Western blot showed that GST-MBP-STYXL1 protein existed in the supernatant as a soluble protein in the *E. coli* expression systems (**Fig. R4**). And GST-MBP-STYXL1 was used instead of GST-STYXL1 in the GST pull-down assay. For the GST pull-down assay, the mouse GST-MPB-STYXL1 protein was purified and immobilized on beads and then incubated with testis lysate. GST-MBP protein was used as a negative control. The eluates were subjected to SDS-PAGE for Coomassie blue staining. As suggested by the reviewer, we showed a gel with four lanes: GST alone, GST-STYXL1 alone, GST with lysate, GST-STYXL1 with lysate (**Supplementary Fig. 5A**). The arrow and arrowhead indicate the bands of GST-MBP protein and GST-MBP-STYXL1 protein, respectively. And several bands unique to GST-STYXL1 with lysate can be observed. Because some testicular proteins pulled down by GST-STYXL1 might be of low amount and beyond the detection limit of Coomassie blue staining, which has low sensitivity, the whole lane of lane 3 and 4 shown in **Supplementary Fig. 5A** were retrieved for trypsin digestion and identification by mass spectrometry. To avoid confusion, “GST-STYXL1” is corrected as “GST-MPB-STYXL1” and “GST” is corrected as “GST-MBP” in the revised manuscript in the **Supplementary Fig. 5A** (Page 9, line 225-228).

Fig R1. plasmid profile: pTXB1-6xHis-Styxl1

Fig R2. Coomassie blue staining for *E. coli* lysate with STYXL1 protein expressed using pTXB1-6xHis-Styx11 plasmid. Red box indicated the band of mouse STYXL1 protein.

Fig R3. plasmid profile: pTXB1-GST-MBP-6xHis-Styx11

Fig R4. Western blot for Styx11 protein expressed in *E. coli* using pTXB1-GST-MBP-6xHis-Styx11 plasmid.

Supplementary Fig. 5A. SDS-PAGE analysis and Coomassie blue staining of the eluates from GST-beads. The arrow and arrowhead indicate the bands of GST-MBP and GST-MBP-STYXL1 protein, respectively. The asterisks indicate the bands that were differentially pulled down by GST-MBP-STYXL1 beads from testis lysate compared with GST-MBP with lysate and GST-MBP-STYXL1 without lysate.

Minor comments:

6) Line 92: "...Eya4 are essential for embryonic lethality". The authors probably meant to write: "...Eya4 are essential to prevent embryonic lethality"?

Response: Thank you for your suggestion. We have rephrased "Eya4 are essential for embryonic lethality." (line 92 in initial manuscript) to "Eya4 is essential to prevent embryonic lethality." (Page 5, line 96) in the revised manuscript.

7) Figure 5C: "Tyrosine" not "Tryosine". You should also change the color scheme for the "helix/sheet-loop-helix/sheet" and replace with a single color for the N-term rhodanese-like domain.

Response: Thanks for this suggestion. We have corrected the spelling error and modified **Fig. 5C** to replace with a single color for the N-term rhodanese-like domain in the revised manuscript.

Fig 5C. Schematic representation of full-length (FL), N-terminal (N, 1-167 aa), C-terminal (C, 168-310 aa) of STYXL1.

8) Line 305: The study cited in reference 51 (Isrie et al) was later disproved by Hengel et al

(2019; Eur J Med Genet). The mutation P311A is likely benign. Thus this mutation in Styxl1 does not cause neurological disorders nor epilepsy. Please correct.

Response: Thanks for this comment. We have updated the cited reference and corrected the description in the revised manuscript (Page 15, line 379).

Reviewer #2 (Remarks to the Author):

Overall, this manuscript adds knowledge on not only phenotype of Styxl1 mouse mutants, but also knowledge on the associated pathomechanisms related to male infertility.

However, three previously published papers already addressed regulation of or regulation by the chaperonin CCT complex during spermiogenesis using mouse models, without being referenced in this manuscript. Here, the authors overlooked the novelty of this study and findings in relation to these previously published studies, and therefore this study lacks a conceptual advance that is a standard in Nature publications:

Li M, Chen Y, Ou J, Huang J, Zhang X. PDCL2 is essential for spermiogenesis and male fertility in mice. *Cell Death Discov.* 2022 Oct 17;8(1):419. doi: 10.1038/s41420-022-01210-2. PMID: 36253364; PMCID: PMC9576706.

Zheng M, Chen X, Cui Y, Li W, Dai H, Yue Q, Zhang H, Zheng Y, Guo X, Zhu H. TULP2, a New RNA-Binding Protein, Is Required for Mouse Spermatid Differentiation and Male Fertility. *Front Cell Dev Biol.* 2021 Feb 18;9:623738. doi: 10.3389/fcell.2021.623738. PMID: 33763418; PMCID: PMC7982829.

Yang P, Tang W, Li H, Hua R, Yuan Y, Zhang Y, Zhu Y, Cui Y, Sha J. T-complex protein 1 subunit zeta-2 (CCT6B) deficiency induces murine teratospermia. *PeerJ.* 2021 Jun 1;9:e11545. doi: 10.7717/peerj.11545. PMID: 34141486; PMCID: PMC8176918.

Response: We appreciate for these constructive suggestions by the reviewer. As shown in the previously published papers, *Pdcl2*^{-/-} mice displayed malformed and immotile sperm. PDCL2 can interact with CCT and actin, based on which the authors speculated that PDCL2 might regulate actin production by CCT. However, actin folding was not studied and the function of CCT wasn't evaluated either. Deletion of *Tulp2* caused defective sperm tail structures and TULP2 interacted with CCT8. The authors speculated that TULP2 might be the substrate of CCT8 and be correctly folded by the CCT complex, however, they didn't provide experimental evidence to confirm TULP2 as a substrate of CCT complex. *Cct6b*^{-/-} mice showed normal spermatogenesis with normal sperm motility and male fertility. Mild increase in ratio of sperm nuclear base bending was observed. These data indicated that the testis-specific protein CCT6B is not essential for murine spermatogenesis. It is possible that CCT6B is not important for CCT complex assembly or spermatogenesis, and CCT6A might play major regulatory roles in spermatogenesis. Thus, the regulation of CCT complex in sperm flagella formation still remains not well known. Here, in our study, we found STYXL1 as a new CCT complex regulator. STYXL1 is essential for the formation of the intact CCT complex during spermiogenesis, working together to fold tubulin and facilitate microtubule polymerization in sperm axonemal formation. We have added the above analysis of literatures mentioned by the reviewer in the Discussion section of the revised manuscript (Page 15, line 379-397).

Major comments:

1) Figure 1 A and B: In the text and Figure 1 A, 23 testis-enriched phosphatases are mentioned, but in the heatmap (Figure 1B) only 22 are presented. What happened to number 23?

Response: Thank the reviewer for the comment. We overlapped phosphatases and testis-enriched genes using their **Ensemble gene IDs**, and among 233 phosphatases, 23 were testis-enriched. When analyzing their expression profile, we compared these 23 testis-enriched phosphatases and da Cruz et al.'s expression data of testicular cells by **gene symbols**, PUDP (pseudouridine 5'-phosphatase) showed missing expression value, that's why only 22 proteins are presented with expression data in the heatmap in Fig. 1B. However, during revision we performed literature analysis, and found that Pudp was also known as HDHD1A. da Cruz et al.'s expression data of testicular cells included HDHD1A, and it has high expression levels in round spermatids. Previous study has shown that HDHD1A might be central in the epigenetic and genomic regulation of sex chromosome aneuploidies (Viuff M et al 2023 *Genome Med*; PMID: 36978128). We have corrected the heatmap in **Fig. 1B** by adding the expression data of HDHD1A in the heatmap in **Fig. 1B** and literature information of HDHD1A in the revised manuscript (Page 5, line 99-100).

Fig 1B. Heatmap of expression levels of testis-enriched phosphatases in different developmental cells. 2C: somatic cells and spermatogonia, LZ: leptotene/zygotene spermatocytes, PS: pachytene spermatocytes, RS: round spermatids.

C

Gene	Family	Phenotype
PPM1B	PPM	embryonic lethality
STYXL1	DSP	-
PGP	NagD	embryonic lethality
EYA4	EYA	die shortly after birth
CDC14A	DSP	spermatogenic defect
PPEF1	PPP	sperm fertilizing and motility
DUSP21	DSP	-
DUSP13	DSP	meiosis regulation
PP2D1	PPM	dispensable for spermatogenesis
HDHD1A	HHDC	regulation of sex chromosome aneuploidies

Fig 1C. The phosphatase family and phenotypes of proteins from Cluster C2 with highest expression in round spermatids.

2) Figure 1 D: it is surprising that *Styx11* mRNA was only detectable in testis, using RT-PCR. According to publicly available data, *STYXL1* shows similar expression in other tissues compared to testis. In addition, publicly available antibody stainings also demonstrate *STYXL1* localization in diverse tissues, not only testis. Please explain this discrepancy and/or provide additional data. Overall, the (q)RT-PCR results, also those presented in Figure S1, are not convincing.

Response: Thank the reviewer for the comments. According to the NCBI Mouse ENCODE transcriptome data (Yue et al 2014 *Nature*; PMID: 25409824) (<https://www.ncbi.nlm.nih.gov/gene/76571>), *Styx11* mRNA is enriched in testis among the mouse thirty tissue samples (**Fig. R5**). In the BioGPS Dataset, (Wu et al 2009 *Genome Biol*, PMID: 19919682; Wu et al 2016 *Nucleic Acids Res*, PMID: 26578587) (<http://biogps.org/#goto=genereport&id=76571>), *Styx11* shows predominant expression in testis compared to other multi-tissues and cell types (**Fig. R6**). Furthermore, according to the Li et al.'s expression profile of mouse tissues (Li et al 2017 *Sci Rep*; PMID: 28646208), *Styx11* mRNA is also predominantly expressed in the testis (**Fig. R7**), which can be clearly visualized using the website (<https://expressionviewer.shinyapps.io/version1/>). And only trace amount of expression is detected in lung and ovary. As RT-PCR in Fig. 1D of initial version of the manuscript might not be sensitive enough to detect trace amount of expression, we performed qRT-PCR (**Fig. 1D** in the revised manuscript) during revision, and confirmed the predominant expression in testis with trace amount of expression in lung, which is consistent with the above data from public databases.

Mouse ENCODE transcriptome data

See details

- Project title: Mouse ENCODE transcriptome data
- Description: RNA profiling data sets generated by the Mouse ENCODE project.
- BioProject: PRJNA66167
- Publication: PMID 25409824
- Analysis date: n/a

Fig R5. The expression levels of *Styx11* mRNA in NCBI Mouse ENCODE transcriptome data

Fig R6. The expression levels of *Styx11* mRNA in BioGPS Dataset.

symbol:
 Styxl1
 gene Id:
 ENSMUSG00000019178
 histogram of expression:

Fig R7. The expression levels of *Styx11* mRNA in Li et al.'s expression profile of mouse tissues.

Fig 1D. qRT-PCR analysis of expression levels of *Styx11* mRNA in eight mouse tissue. 18S rRNA was used as a loading control.

3) Figure 2 A: The authors should provide evidence by Western blot analysis or immunofluorescence analysis, that STYXL1 is lacking in mutant mice. Several antibodies are commercially available that could have been used for this purpose and a full knock out mouse model should demonstrate complete absence of this protein.

Response: We appreciate for this comment. Actually, we have already tried the commercial anti-STYXL1 antibody (NOVUS BIOLOGICALS brand, NBP2-93417, Rabbit polyclonal antibody), and subsequently generated custom-made rabbit STYXL1 antibody in Abclonal (Wuhan, China). We performed Western blot and immunofluorescence analysis using the two antibodies (**Fig. R8 and R9**). Unfortunately, the bands shown in the Western blots were non-specific and we didn't get any positive signal in immunofluorescence analysis. Thus, these antibodies are unable to detect STYXL1 protein in Western blot and immunofluorescence analysis. To find out whether STYXL1 protein is lacking in mutant mice, relative protein quantification of STYXL1 in testis was measured using targeted quantification based on a parallel reaction monitoring (PRM) method by LC-MS/MS. The results showed that testes from both two mutant lines of *Styx11*^{Δ74bp/Δ74bp} and *Styx11*^{Δ4bp/Δ4bp} showed absence of STYXL1 protein

expression (**Fig. 2B and Supplementary Fig. 6**), indicating the successful deletion of STYXL1 and generation of *Styxl1*^{-/-} mouse. We have added above results in the revised manuscript (Page 6, Line 127-133).

Fig R8. Western blot analysis of STYXL1 protein in *Styxl1*^{+/+} and *Styxl1*^{-/-} testes using commercial and custom-made anti-STYXL1 antibodies

Fig R9. Immunofluorescence analysis of *Styxl1*^{+/+} and *Styxl1*^{-/-} testes using commercial and custom-made anti-STYXL1 antibodies (green). Nuclei were stained with DAPI. Scale bar: 20µm.

Fig 2B. Quantification of relative expression levels of STYXL1 protein in *Styxl1*^{Δ74bp/Δ74bp} and *Styxl1*^{Δ4bp/Δ4bp} testes by PRM. N=3. Data were presented with the mean ± SEM. ***, p < 0.001.

Supplementary Fig 6. Representative extracted ion chromatograms of three peptides from STYXL1 protein in quantification measurements by PRM. Heavy peptides with BSA were used as a negative control.

4) Figure 3A and B: it is difficult to appreciate the differences between the control and mutant animals.

Response: Thank you for this comment. Spermiogenesis involved in different developmental steps from step 1 to step 16 based on the shapes of the acrosome of spermatids and stages of seminiferous tubules. As shown in **Fig. 3A**, wildtype testis had sperm flagella arranged in the lumen at stage VII-VIII, while in *Styx11*^{-/-} testis few flagella were observed in the lumen at stage VII-VIII (as indicated by asterisk). Since step 10, the nucleus of spermatids progressively elongates and finally transformed into sickle shaped nucleus in wildtype testis (step 16). While in *Styx11*^{-/-} testis, nucleus of spermatids was abnormally elongated, and don't show classical sickle shape (as indicated by the red arrows). To better show the phenotypes, the zoom-in views of *Styx11*^{-/-} malformed elongating spermatids and corresponding wild type controls were shown in **Fig. 3B** together with their schematic diagrams.

Fig 3A-B. (A) Different stages of seminiferous tubules in PAS-stained *Styx11*^{+/+} and *Styx11*^{-/-} testes. The red arrow indicates elongating spermatids with abnormal nuclei. Asterisk indicated sperm tails with defects. L, leptotene; Z, zygotene; P, pachytene; D, diplotene; RS, round spermatid; ES, elongated spermatid. Scale bar: 25µm. (B) Enlarged pictures of different steps of *Styx11*^{+/+} and *Styx11*^{-/-} elongated spermatids pointed by arrows in (A) were presented. Schematic diagrams were denoted at bottom.

5) Figure 6 B, line 240: provided images do not convincingly demonstrate that loss of STYXL1 affects distribution of CCT1, CCT6, CCT7 in sperm flagella. I recommend to exchange these images showing STYXL1 deficient sperm with severe MMAF with images of less malformed sperm flagella. These should be available as suitable examples have been shown in Figure 2G and 3B (top). In addition, despite low sperm count compared to control (Figure 2B), only about 35-40% of those present with morphological abnormalities, therefore a more convincing IF analysis can and should be provided.

Response: We appreciate for the suggestion by the reviewer. As suggested by the reviewer, we have provided images of sperm having less malformed flagella with immunofluorescent staining of CCT1, CCT6 and CCT7. We found that CCT1, CCT6 and CCT7 remained restricted

to the sperm flagella but with discontinuous distribution after deletion of *Styx11* (**Fig. 6B**). Further Western blot analysis indicated that the expression levels of CCT1 significantly decreased in *Styx11*^{-/-} sperm (**Fig. 6C**). The above immunofluorescent results have been updated in the revised manuscript (Page 11, line 266-268).

Fig 6B. Immunofluorescence analysis of AC-TUBULIN (green) and CCT1, CCT6 or CCT7 (yellow) in *Styx11*^{+/+} and *Styx11*^{-/-} sperm with nuclei stained by DAPI (blue). Scale bar: 10µm.

Fig 6C. Western blot and quantitative analysis of CCT1, CCT6 and CCT7 in sperm lysate from *Styx11*^{+/+} and *Styx11*^{-/-} mice. PRM2 was used as a loading control. N=3. Data were presented as mean ± SEM. NS, not significant; *, $p < 0.05$; **, $p < 0.01$; ***, $p < 0.001$.

6) Line 121: it remains unclear, if the 4 bp and 74 bp deletion is on one the same or different alleles. If two independent mutant lines have been generated, which of those has undergone deeper analysis and if both have been analysed (and only results of one strain are presented) do they show the same results?

Response: Thank you for your comments. To examine the function of STYXL1 in spermatogenesis, we generated *Styx11* knockout mice using CRISPR-Cas9 system and obtained two independent mutant lines with one having 4bp and the other having 74bp deletion. Because commercial anti-STYXL1 antibody (NOVUS BIOLOGICALS brand, NBP2-93417, Rabbit polyclonal antibody) and custom-made rabbit STYXL1 antibody by Abclonal (Wuhan, China) couldn't detect STYXL1 protein by Western blot or immunolocalization analysis, we performed targeted quantification analysis of STYXL1 protein based on a parallel reaction monitoring (PRM) method by LC-MS/MS, and found that STYXL1 protein is absent in the testes of both two mutant lines (**Fig. 2B**). Many phenotype data in our original manuscript during initial submission were based on both two mutant lines and shown together with no distinction. In the revised manuscript, we separated the results of two mutant lines including testis/body ratio,

fertility test, sperm parameter and morphological analysis, and found that these two mutant lines show the same phenotypes (**Fig. 2C-J, Fig. 3A-B, Supplementary Fig. 2C-I and Supplementary Fig. 3A**). As *Styx11*^{Δ74bp/Δ74bp} mice were used for in-depth studies, the results of *Styx11*^{Δ74bp/Δ74bp} mice were mainly presented in the main manuscript and presented as *Styx11*^{-/-} mice, while the results of *Styx11*^{Δ4bp/Δ4bp} mice were mainly presented in supplementary files. We have added above results in the revised manuscript (Page 6, line 124-133).

Fig 2B. Quantification of relative expression levels of STYXL1 protein in *Styx11*^{Δ74bp/Δ74bp} and *Styx11*^{Δ4bp/Δ4bp} testes by PRM. N=3. Data were presented with the mean ± SEM. ***, *p* < 0.001.

Fig 2C-J. (C) Litter size of *Styx11*^{-/-} male and *Styx11*^{-/-} female mice. N=5. (D) Statistics analysis

of *Styx11*^{+/+} and *Styx11*^{-/-} testis/body weight ratio. N=3. (E-G) Quantitative analysis of sperm count (E), sperm motility (F) and progressive motility (G) of adult *Styx11*^{+/+} and *Styx11*^{-/-} mice. N=4. (H) H&E-stained caput epididymis and cauda epididymis of *Styx11*^{+/+} and *Styx11*^{-/-} mice. Scale bar:50μm. (I-J) The morphologies of *Styx11*^{-/-} sperm by H&E staining (I) and the percentage of sperm abnormalities (J) in *Styx11*^{-/-} cauda epididymis compared with controls. Scale bar:5μm. N=3. Data were presented with the mean ± SEM. NS, not significant; **, *p* < 0.01; ***, *p* < 0.001.

Supplementary Fig 2C-I. (C) Litter sizes of *Styx11*^{Δ4bp/Δ4bp} male and *Styx11*^{Δ4bp/Δ4bp} female mice. (D) Testis/body weight ratios of adult *Styx11*^{Δ4bp/Δ4bp} and control mice. (E-G) Quantitative analysis of sperm count (E), sperm motility (F) and progressive motility (G) of adult *Styx11*^{+/+} and *Styx11*^{Δ4bp/Δ4bp} mice. (H-I) The morphologies of *Styx11*^{Δ4bp/Δ4bp} sperm by H&E staining (H) and the percentage of sperm abnormalities (I). Scale bar:5μm. N=2. Data were presented as the mean ± SEM. NS, not significant; *, *p* < 0.05; **, *p* < 0.01; ***, *p* < 0.001.

Fig 3A-B. (A) Different stages of seminiferous tubules in PAS-stained *Styx11*^{+/+} and *Styx11*^{-/-} testes. The red arrow indicates elongating spermatids with abnormal nuclei. Asterisk indicated sperm tails with defects. L, leptotene; Z, zygotene; P, pachytene; D, diplotene; RS, round spermatid; ES, elongated spermatid. Scale bar: 25µm. (B) Enlarged pictures of different steps of *Styx11*^{+/+} and *Styx11*^{-/-} elongated spermatids pointed by arrows in (A) were presented. Schematic diagrams were denoted at bottom.

Supplementary Fig 3A-B. (A) Different stages of seminiferous tubules in PAS-stained *Styx11*^{+/+} and *Styx11*^{Δ4bp/Δ4bp} testes. The red arrows indicate elongating spermatids with abnormal nuclei. Asterisk indicates defects of sperm tails. L, leptotene; Z, zygotene; P, pachytene; D, diplotene; RS, round spermatid; ES, elongated spermatid. M, metaphase. Scale bar: 25µm. (B) Enlarged pictures of different steps of *Styx11*^{+/+} and *Styx11*^{Δ4bp/Δ4bp} elongated spermatids pointed by arrows in (A) were presented. Schematic diagrams were denoted at bottom.

7) Line 127: The authors should provide videos from wildtype control and *Styx11*^{-/-} sperm to provide additional evidence.

Response: Thanks for the reviewer's suggestions. We have provided the video (Supplementary Movie 1) recording of sperm from *Styx11*^{+/+}, *Styx11*^{Δ74bp/Δ74bp}, *Styx11*^{Δ4bp/Δ4bp} mice in the revised manuscript.

8) Line 174 following: please explain, why knockdown in NIH3T3 cells was used to assess the impact of STYXL1 on cilia in fibroblasts instead of taking fibroblasts from their established mouse line using standard protocols? Also, fibroblasts do not represent the right "test tube" to assess function of genes that are potentially involved in motile cilia generation or sperm cell

maturation.

Response: Thank the reviewer for the comments. Both cilia (motile and non-motile cilia) extend from a basal body that consists of triplet microtubules, and subdistal and distal appendages (Reiter JF et al 2017 *Nat Rev Mol Cell Biol*; PMID: 28698599). Axonemes which are composed of doublet microtubules are the core elements of cilia. In sperm flagellum, the motile cilia, axonemes usually contain '9+2' peripheral doublet microtubules and central microtubule pair (CP) surrounded by accessory structures that are required for sperm motility. Primary cilia, the non-motile cilia, also comprise peripheral doublet microtubules but lack a central pair of microtubules. Since there is no ideal *in vitro* model to study the function of sperm flagellum so far, NIH3T3 cells, which can be used to induce primary cilia after serum starvation, are widely used instead in many studies of sperm flagella formation by many laboratories (Zhu et al 2023 *Elife* PMID: 36756949; Zhao et al 2018 *J Cell Mol Med*, PMID: 29168316; Crapster JA et al 2020 *Elife* PMID: 32163033; Nishimura N et al 2008 *Arch Biochem Biophys*, PMID: 18396146). Since STYXL1 and CCT complex subunits are endogenously expressed in NIH3T3 cells, we found that after serum starvation, *Styx11* knockdown also greatly decreased the length of cilia in NIH3T3 cell. And deletion of *Styx11* in mice led to defects in sperm flagella, the motile cilia (absent, short, coiled, bent or irregular width). Both *in vitro* and *in vivo* results indicate that STYXL1 is important for cilia formation. We found that the vast majority of α -TUBULIN and β -TUBULIN was able to pellet under polymerization conditions in control NIH3T3 cells, in contrast, in NIH3T3 cells with *Styx11* knockdown, α -TUBULIN and β -TUBULIN remained in the supernatant and are unable to polymerize. The MT sedimentation assay further showed that *Styx11* deletion led to defects in tubulin polymerization in sperm. Both *in vitro* and *in vivo* results indicate that STYXL1 plays an important role in tubulin polymerization and microtubule organization. Above results indicated that NIH3T3 cells can be well used as an *in vitro* model to study the function of STYXL1 in CCT-based tubulin folding.

9) Line 350: Sperm motility defects in *Styx11* mutants are likely caused by MMAF. Here, the conclusion that STYXL1 is not only regulating sperm flagella formation but also sperm motility appears not appropriate, based on data provided.

Response: We appreciate for the suggestion by the reviewer. We agree with the reviewer that sperm with flagella formation abnormalities have motility defects. We have rephrased "germ cell-specific STYXL1 is essential in the formation of sperm flagella and the regulation of sperm motility." (line in initial manuscript) to "germ cell-specific STYXL1 is essential in the regulation of sperm flagella formation" (Page 15, line 399-400) in the revised manuscript.

10) Proteomics: The interaction data, especially regarding interaction with CCT1, CCT6 and CCT7 are convincing. However, regarding the LC-MS/MS approach, the threshold for differential expression (2 fold) appears to be very low. Thus, the dataset is likely overestimated. A re-analysis using 3 to 5 fold difference as cutoff should be performed. Also, a great variability between samples is to be expected that could lead to false positive/false negative data due to the low cutoff. Thus, it is in addition important to know how many samples have been analyzed.

Response: Thank the reviewer for the comment. For GST pull-down assay followed by LC-MS/MS analysis, three biological replicated were performed. According to the reviewer's suggestion, we have used the 3 fold difference as a cutoff for differential expression analysis in

the revised manuscript. And CCT complex containing CCT1-CCT8 subunits were still identified. The above criteria have now been added in the revised manuscript (Page 19, line 500-503).

Minor comments:

Complete manuscript:

- please check and use correct capitalization of protein symbols.
- please check for spellings and grammar.

Response: Thank you for this suggestion. We have now used correct capitalization of protein symbols and worked on both language and readability. The spellings and grammar have been carefully checked and corrected.

REVIEWERS' COMMENTS

Reviewer #1 (Remarks to the Author):

The authors have satisfactorily addressed all of my concerns. This is an important and well executed study. I recommend publication.

Reviewer #2 (Remarks to the Author):

The authors have significantly strengthened the manuscript during the revision process with new supporting data and appropriate corrections to the text. This study will add significant knowledge to the field of male fertility and the still poorly understood intricacies of the CCT/tubulin folding complex. I therefore recommend this study for publication in Nature Communications.

Reviewer #1 (Remarks to the Author):

The authors have satisfactorily addressed all of my concerns. This is an important and well executed study. I recommend publication.

Response: We would like to thank the reviewer for the assessments of our work. And the comments have significantly improved our manuscript.

Reviewer #2 (Remarks to the Author):

The authors have significantly strengthened the manuscript during the revision process with new supporting data and appropriate corrections to the text. This study will add significant knowledge to the field of male fertility and the still poorly understood intricacies of the CCT/tubulin folding complex. I therefore recommend this study for publication in Nature Communications.

Response: We thank the reviewer for these assessments and positive comments of our work.